# The use of health geography modeling to understand early dispersion of COVID-19 in São Paulo, Brazil

**Carlos Magno Castelo Branco Fortaleza**[1]◉*, **Raul Borges Guimarães**[2]◉, **Rafael de Castro Catão**[3‡], **Cláudia Pio Ferreira**[4‡], **Gabriel Berg de Almeida**[1‡], **Thomas Nogueira Vilches**[5‡], **Edmur Pugliesi**[2‡]

**1** Department of Infectious Diseases, Botucatu Medical School, São Paulo State University (UNESP), Botucatu, São Paulo State, Brazil, **2** Department of Geography, Faculty of Science and Technology, São Paulo State University (UNESP), Presidente Prudente, São Paulo State, Brazil, **3** Department of Geography, Federal University of Espírito Santo, Vitória, Espírito Santo State, Brazil, **4** Institute of Biosciences, São Paulo State University (UNESP), Botucatu, São Paulo State, Brazil, **5** Institute of Mathematics, Statistics and Scientific Computation, University of Campinas (UNICAMP), Campinas, São Paulo State, Brazil

◉ These authors contributed equally to this work.
‡These authors also contributed equally to this work.
* carlos.fortaleza@unesp.br

**Data Availability Statement:** All relevant data are within the manuscript and its Supporting information files.

## Abstract

Public health policies to contain the spread of COVID-19 rely mainly on non-pharmacological measures. Those measures, especially social distancing, are a challenge for developing countries, such as Brazil. In São Paulo, the most populous state in Brazil (45 million inhabitants), most COVID-19 cases up to April 18th were reported in the Capital and metropolitan area. However, the inner municipalities, where 20 million people live, are also at risk. As governmental authorities discuss the loosening of measures for restricting population mobility, it is urgent to analyze the routes of dispersion of COVID-19 in São Paulo territory. We hypothesize that urban hierarchy is the main responsible for the disease spreading, and we identify the hotspots and the main routes of virus movement from the metropolis to the inner state. In this ecological study, we use geographic models of population mobility to check for patterns for the spread of SARS-CoV-2 infection. We identify two patterns based on surveillance data: one by contiguous diffusion from the capital metropolitan area, and the other hierarchical with long-distance spread through major highways that connects São Paulo city with cities of regional relevance. This knowledge can provide real-time responses to support public health strategies, optimizing the use of resources in order to minimize disease impact on population and economy.

## Introduction

The International Health Regulations (IHR), administered by World Health Organization (WHO), was last revised in 2005, under the influence of the global response to the SARS emergency and the risk of the H5N1 influenza pandemic [1]. Since then, it has guided coordinated

**Funding:** We thank Centro de Vigilância Epidemiológica of São Paulo State (CVE), Health Department, to provide the data. CPF and TNV thank the support from São Paulo Research Foundation (FAPESP) grant 18/24058-1, 18/24811-1, respectively.

**Competing interests:** The authors have declared that no competing interests exist.

international cooperation during public health emergencies such as the Zika virus and Ebola epidemics [2]. However, the current COVID-19 pandemic is the greatest challenge faced by IHR thus far [3]. Although the WHO has issued several guidelines related to the current epidemic, the adherence level varies among nations and, inside nations, provinces, and states [4].

Up to the present day, non-pharmacological interventions, like social distancing, radical lockdown, and extensive testing for SARS-CoV-2 infection, have been applied by different countries, with widely varying degrees of success [5, 6]. In some countries, such as Brazil, scientific research on the effectiveness of those strategies has been severely hampered by political bias, which interferes with public health decisions [7].

São Paulo, the most populous state in Brazil (45 million inhabitants), is also the most severely affected by COVID-19. The state government has challenged the Brazilian President's denial of the pandemic and declared the closure of commerce, schools and other non-essential services. However, despite the ferocious spread of the virus on the state capital and metropolitan area, the slowly evolving of the epidemic in the state's inner cities (until April 18th), where 20 million people live, has led to protests against governmental measures. In this context, there is a sense of urgency about predicting routes of epidemic spreading in the inner state and the population's risks.

Here, we discussed a detailed analysis of the spatial dispersion of COVID-19 in São Paulo State, Brazil, intending to provide real-time responses to support public health strategies. Using data since the first confirmed cases of COVID-19 in São Paulo State, we assess the importance of geographic space on the spread of the epidemic. We hypothesize that urban hierarchy is the main responsible for the disease spreading, and we identify the hotspots and the main routes of virus movement from the metropolis to the inner state. This premise is also supported by [8] where multivariate analyses showed that demographic density and high classification of regional relevance were associated with early introduction and high COVID-19 incidence and mortality rates. We cross validate the confirmed cases with urban mobility, urban hierarchy, and land use at each spatial localization, in work developed here. The results highlight the importance of the main routes that cross São Paulo State and the regional airports on introducing the disease in the territory, just as the main municipalities act as critical centers of disease spreading to the inner state. Knowing in advance the path of COVID-19 dispersion can support decision-makers to optimize health service, and plan strategies of quarantine measures. This approach can be made in other states of Brazil and other developing countries, observing local and regional mobility and urban network [9].

## Methods

### Geographical data modelling

Spatial analysis of surveillance data includes exploratory data analysis, spatial modeling, and cartographic visualization [10]. The first one uses spatial statistical methods to measure centrality and dispersion of data sets to detect spatial patterns and check for relationships between variables of the complex phenomenon under investigation. The second one examines the elementary forms of spatial organization that explains the phenomenon under study, such as railways, land cover, demographic, economic, and social factors [11]. Lastly, cartographic visualization provides a synthesis of the previous procedures, aiming the elaboration of a thematic map that can be presented to managers for decision making in emergencies in public health [9, 10].

Focusing on São Paulo State, its center and periphery structure, main roads, and network flux that gives population and trade mobility, the geographic spread of COVID-19 was studied. For this, several maps were made to summarize information about quantity and localization of

confirmed cases, urban hierarchy, area of influence and urban-rural typology of cities, modes of transport, and population vulnerability. The maps were constructed based on graphic-semiology principles, the theory of colors and visual communication [10, 12, 13].

We used surveillance data (number of confirmed cases of coronavirus) updated on April 18th, 2020. The data was obtained from Brasil IO's compiled databank (https://brasil.io/dataset/covid19/boletim/) kept by volunteers' task force (researchers and reporters). This group daily catches, from the epidemiological reports of each state, the number of confirmed cases and death by SARS-CoV-2 and make it publicly available. Because of the recognition of SARS-CoV-2 as a pandemic by the WHO, laboratory certification in Brazil ranged from few laboratories to 26, in eight weeks since the beginning of the epidemic; the majority is located at São Paulo State [14]. Data reporting of Severe Acute Respiratory Illness (SARI) is mandatory in Brazil. A specific form (national database SIVEP-Gripe) collects information that allows us to estimate reported delay, disease fatality at which age class, and identify confirmed cases of the disease. This permits surveillance of all respiratory diseases in Brazil. Only cases that were hospitalized belong to this data set; therefore, underreporting is expected. On the other hand, this is probably homogeneous along with the municipalities and will not impact on the observed pattern of disease spreading. Out of this national surveillance system (SARI), test capacity can vary among cities because many of them made agreements with factories to test the population that lives on the site where the factory is settled. The same procedure has been done in universities, schools, and firms that returned their activities. However, this data has not been taken into account here, since SARI only reports severe cases. Lastly, test capacity has grown fastly in São Paulo State; currently, 13 out of 35 laboratories of Brazil are settled in this state. A broader serological survey is on course in Brazil, to detect underreporting and follow the population susceptibility along with the course of the epidemic. This may help in defining target groups for vaccination.

Data about each municipality, such as territorial management, trade and services, financial services, health care services, educational institutions, media and communication markets, culture and sport, mode of transport, and land use, was used to identify the fundamental entities of spatial structure that trigger coronavirus dispersion in São Paulo territory [15]. This information was compiled from census data done by the Federal Government and other thematic studies. The metropolis of São Paulo appears as the largest urban complex in the country, with almost 22 million inhabitants and a high level of integration with other municipalities that comprise the national territory. It is listed as an alpha global city by the Globalization and World Cities Research Network (GaWC). In the second level of the hierarchy, we have Rio de Janeiro and Brasília (the capital of Brazil). Focusing on São Paulo State, it has two main axes of urban and trade mobility (roadways, railways, and airways), the first one connecting São Paulo with Rio de Janeiro, and the second one connecting São Paulo with Brasília and Central-West Region of Brazil. Besides these main transportation axes, we have a secondary flux network connecting the metropolis of São Paulo to country municipalities and the South of Brazil. This secondary flux aggregates roadways, railways, airways, and waterways (Tietê-Paraná). Over this intense flux of people and trade, a complex structure of cities emerges, reinforcing this network composed of high hierarchy cities (as nodes) linked by the best transportation system of the country (as edges). With almost 48 million inhabitants, São Paulo State concentrates 23.6% of the country's population and 33% of its income. Besides, São Paulo State has the highest number of primary (Metropolis) 2/15, and secondary (Regional Capital) 26/97 cities on the urban hierarchy level in Brazil, and a high number of other cities classified as Subregional Center 77/352, and Zone Center 51/398, respectively, at tertiary and quaternary levels.

Out of 645 municipalities in São Paulo State, 145 have laboratory-confirmed cases on April 18th and were used in this study. The first confirmed case was at São Paulo metropolis on

March 25th. In all maps that we will present, the studied feature (number of coronavirus cases or time-lapse since the first case) is located at each municipality's city hall.

In the first map, we plotted the number of laboratory-confirmed cases reported from March 25th to April 18th. For this, the proportional symbol maps scale was used to draw circles proportionally to the number of cases in each municipality. Proportional symbol maps are often constructed by beginning with the largest symbol size (the largest radius of the circle corresponds to the largest data value) to minimize symbol overlap. To measure the spatial trend on data, we use a weighted standard deviation ellipse. In this case, data of each municipality $i$ was (until the date at which each ellipse was drawn), centered at the city hall (position coordinate ($x_i$, $y_i$)) and weighted by the number of cases in the municipality [16].

Three ellipses were drawn to show, at different times, the main direction of disease spreading. Although the SARS-CoV-2 was introduced in São Paulo on March 25th, it took time to move towards the inner municipalities because of the strong mitigations strategy adopted by São Paulo State to halting the disease's spread. The average time spent by the disease, since its introduction on the metropolis, to achieve the regional centers, the municipalities under major and minor influence, and the rural municipalities were respectively 22, 31, 34 and 55 days [8] (the classification of the municipalities follows the criteria established by the Brazilian Institute for Geography and Statistics (2017) [17]). Therefore, three calendar date were chosen to cover the period of study (from March 25 until April 18): March 29, April 8, and April 18; which are 10 days apart from each other. Over this information, we highlight the main roads that cross São Paulo State, and its 645 municipalities' urban-rural typology.

The standard deviations for the x- and y-axis are given by

$$\sigma_{1,2} = \sqrt{\frac{\sum_i \tilde{x}_i^2 + \sum_i \tilde{y}_i^2 \pm \sqrt{\sum_i \tilde{x}_i^2 - \sum_i \tilde{y}_i^2 + 4\left(\sum_i \tilde{x}_i \tilde{y}_i\right)^2}}{2n}}$$

where

$$\bar{x} = \sum_i x_i, \quad \bar{y} = \sum_i y_i, \quad \tilde{x}_i = x_i - \bar{x}, \quad \tilde{y}_i = y_1 - \bar{y},$$

and the summation symbol $i$ takes into account the number of municipalities with registered cases of COVID-19. Observe that ($\bar{x}$, $\bar{y}$) represents the mean center of the feature. A Standard deviation ellipse summarizes both the dispersion and orientation of the observed set of samples. If the data is normally distributed, one standard deviation represents approximately 68% of all occurrences.

The second map shows the movement of airplanes during March and April of 2020, connecting municipalities of São Paulo State among them and with other states and countries. The data were obtained from [18]. Line thickness is proportional to the number of passengers moving from one place to another. For the purpose of the study, the flux inside São Paulo territory is highlighted.

The third map shows the urban hierarchy centrality level of municipalities. The regional importance of each city can also be seen in this figure from the tree diagram. The data were obtained from [15]. According to the literature, five urban hierarchy levels are defined: metropolis, regional capital, subregional center, zone centers, and local center. Many variables are used in this classification, such as services establishments, inter-urban relations, banking establishments, social information, cultural and sports offering, and territorial management [15, 19].

The fourth map was constructed by interpolating over the total number of days at which coronavirus transmission was reported in each municipality. We used Inverse Distance Weighting (IDW) as an interpolator and a circle as the neighborhood shape for the interpolation procedure. The Root Squared Error (RMS) permitted to set up the radius of the circle (25 km) and the minimum and maximum numbers of neighbors, respectively 2 and 12, to optimize global accuracy of the interpolated curve. Inside of this radius, the nearest neighbors (with reported cases) of each point $s_0$ were used in the interpolator and the contribution of each one was weighted by the inverse of its distance. This gave us an RMS of 6.56. Assuming that the measured values closest to the prediction location have more influence on the predicted value than those far away, the following equation was used

$$\hat{z}(s_0) = \sum_i w_i z(s_i),$$

where $\hat{z}(s_0)$, $w_i$, and $z(s_i)$ are the estimated value at position $s_0$, the weight attributed to each pair of coordinates $(1/|s_i - s_0|)$ and the numerical value observed at position $s_i$. In the summation symbol, $i$ takes into account the number of neighbors.

The interpolator created a surface on which the values from points (municipalities) are combined and recorded in a data matrix, simplifying information, and creating regional patterns. As it has spatiotemporal data, it must be read with the darkest data in the red palette as the oldest that passes through orange, yellow going to the blue palette, which are the municipalities that were later infected. Although we are using the time-lapse since the first case reported in each municipality to create the interpolated surface that permits us to predict the epidemic course over the state, some cities entered community transmission only after 10 to 15 days from the first confirmed case. This reflects not only the stochastic nature of the introduction of a new pathogen in a community, but also the fact that data of COVID-19 in Brazil do not distinguish between imported and autochthonous cases. This is an exploratory analysis that permits us to follow the virus's dispersion pattern, glimpsing the next cities that will probably be affected by the disease. The palette of colours of the map comprises the period from February 25th to April 18th, with white color indicating disease absence.

The last map is a schematic cartogram of the elementary spatial structures that drive and modulate disease spreading in São Paulo territorial. It shows the main modes of transportation, together with the key municipalities that acted as agents of the initial spreading of COVID-19. It also highlights the geographic position of the metropolis and the vulnerable population.

Finally, we want to emphasize that we are not looking for epidemiological links that explain disease transmission among municipalities, but we seek geographical links that conditionate the regional pattern of disease spreading along São Paulo territorial.

## Results and discussion

Fig 1 shows on grayscale the 645 municipalities painted according to their classification of urban-rural typology that takes into account population density, accessibility to goods and services, and land use; São Paulo State has 13% of its area classified as rural [17, 20]. In pink, we highlighted the main roads that cross the state, dividing them into primary and secondary axes according to the flux of people and goods. The municipalities with reported COVID-19 cases are shown in red circles, which size is proportional to the number of cases recorded until April 18th. Three weighted standard deviation ellipses are shown on March 29th, April 08th, and April 18th. The angles are 128, 135, and 137 degrees, respectively, and the semi-major axis measures 34, 89, and 110 km. As time passes and the epidemic evolves, we can notice a change

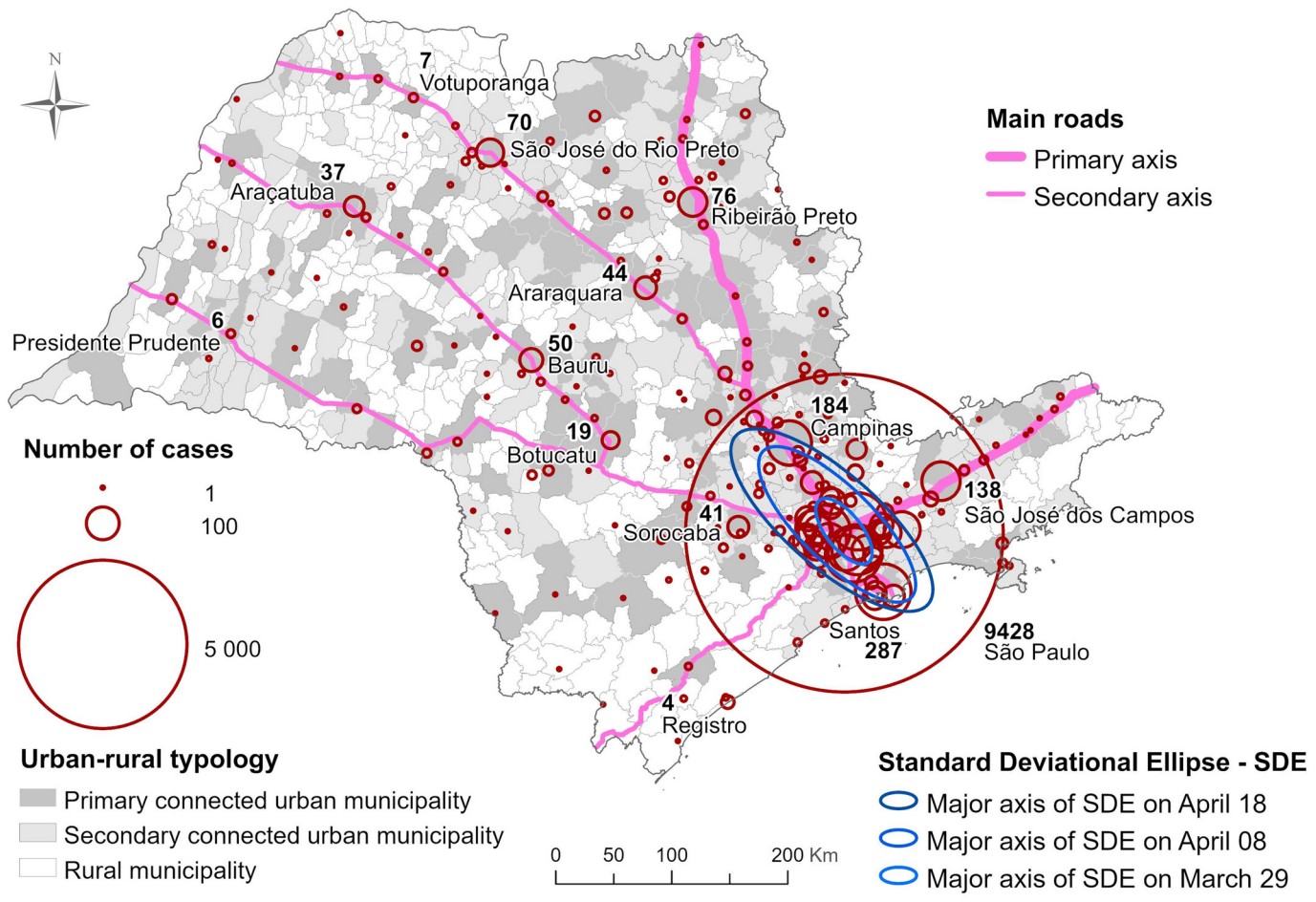

**Fig 1. Distribution of confirmed COVID-19 cases in São Paulo State as of April 18th 2020, Brazil.** The size of the circles is proportional to the number of cases reported in each municipality. The map also shows the main roads that cross the state, the typology of each municipality, and the direction of disease spread on three different moments of epidemic course. The map was made using the software ArcGIS (version 10.8).

of direction and velocity of disease spreading. A simple calculus gives us 5.5 km/day and 2.1 km/day ($\Delta S/\Delta t$ where $\Delta S$ is the difference between the semi-major axis measure and $\Delta t$ is the time elapsed between each ellipse). Interesting to note that on March 24th, mobility restriction was imposed at São Paulo State. Planned to be finished on April 07th, this restriction mobility was extended several times until May 27th when São Paulo quarantine plan started [21].

To emphasize the mobility restrictions imposed by São Paulo government and how connected are its municipalities, Fig 2 shows the airplanes moving in and out of São Paulo during March and April months of 2020. It was registered a movement of 2 to 11350 individuals per connection (107 different connections among different cities) using this option as transportation in March, and 2 to 4 individuals per connection in April (16 connections among different cities), considering that every flight has at least two individuals. In the figure, line thickness is proportional to the number of individuals moving among municipalities. The reduced number in April reflects the travel restrictions imposed by the government of São Paulo to reducing coronavirus spread in the state. We can see that the inner state is well connected not only by roadways (as shown in Fig 1) but also through airways. A highlight to cities of Campinas and São Paulo that have international airports. In order to compare, in March and April of 2019, the number of passengers that left São Paulo State was three times greater than in 2020; and

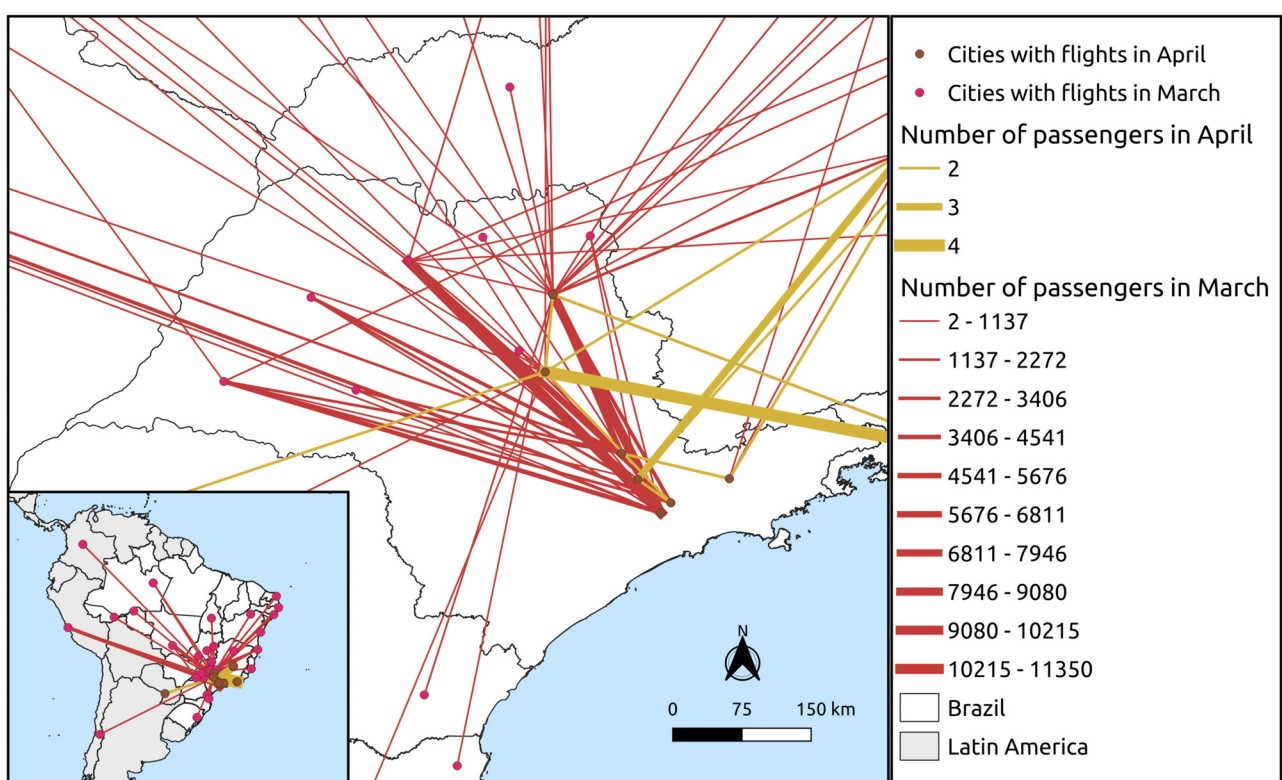

**Fig 2. Airway connections in March and April of 2020 at São Paulo State, Brazil.** Red (March month) and orange (April month) colors show airport connections among different cities of São Paulo, as well as among São Paulo State and other states in Brazil or other countries. Line thickness is proportional to the number of passengers moving from one city to another. The map was made using the software QGIS 3.10.

the number of passengers arriving at São Paulo State's airport was 15 times in 2019 than in 2020.

People's movement is facilitated, and encouraged, due to transportation availability and commercial and social activities Likely SARS-CoV in 2003, the SARS-CoV-2 fastly spread among cities and countries due to airline network and ground transportation [22–24]. In the case of São Paulo State, the delay in closing the airports located at the inner municipalities probably contributed to the hierarchical dispersion of the disease on its territorial.

Fig 3 displays the cities with a high level of urban hierarchy centrality that we can find at São Paulo State: the metropolis of São Paulo, the regional capital of Campinas, the subregional centers of São José do Rio Preto and Ribeirão Preto, the zone centers of Presidente Prudente, Marília, Bauru, Araçatuba, Sorocaba, São José dos Campos, Santos, Araraquara, and Piracicaba, and local center of Barretos, Franca, São João da Boa Vista, São Carlos, Rio Claro, Limeira, Ourinhos, Botucatu, Jaú, and Catanduva. The black border delimits the regions subordinate to the cities level one and two in the hierarchy, and the yellow one the regions subordinate to cities of level three. The diagram on tree summarizes the regions of influence of each city displayed on the map. We hypothesize that city hierarchy plays an important role in the disease spreading over the territory.

The exploratory analysis of data on confirmed cases in São Paulo State generated a dispersion map in which the color spectrum indicates the areas ranging from earlier to the more recent introduction of SARS-CoV-2 (Fig 4). The colors have to be reading such as a predictor of an earlier or later arrival of the disease in each city of the map because they comprise only

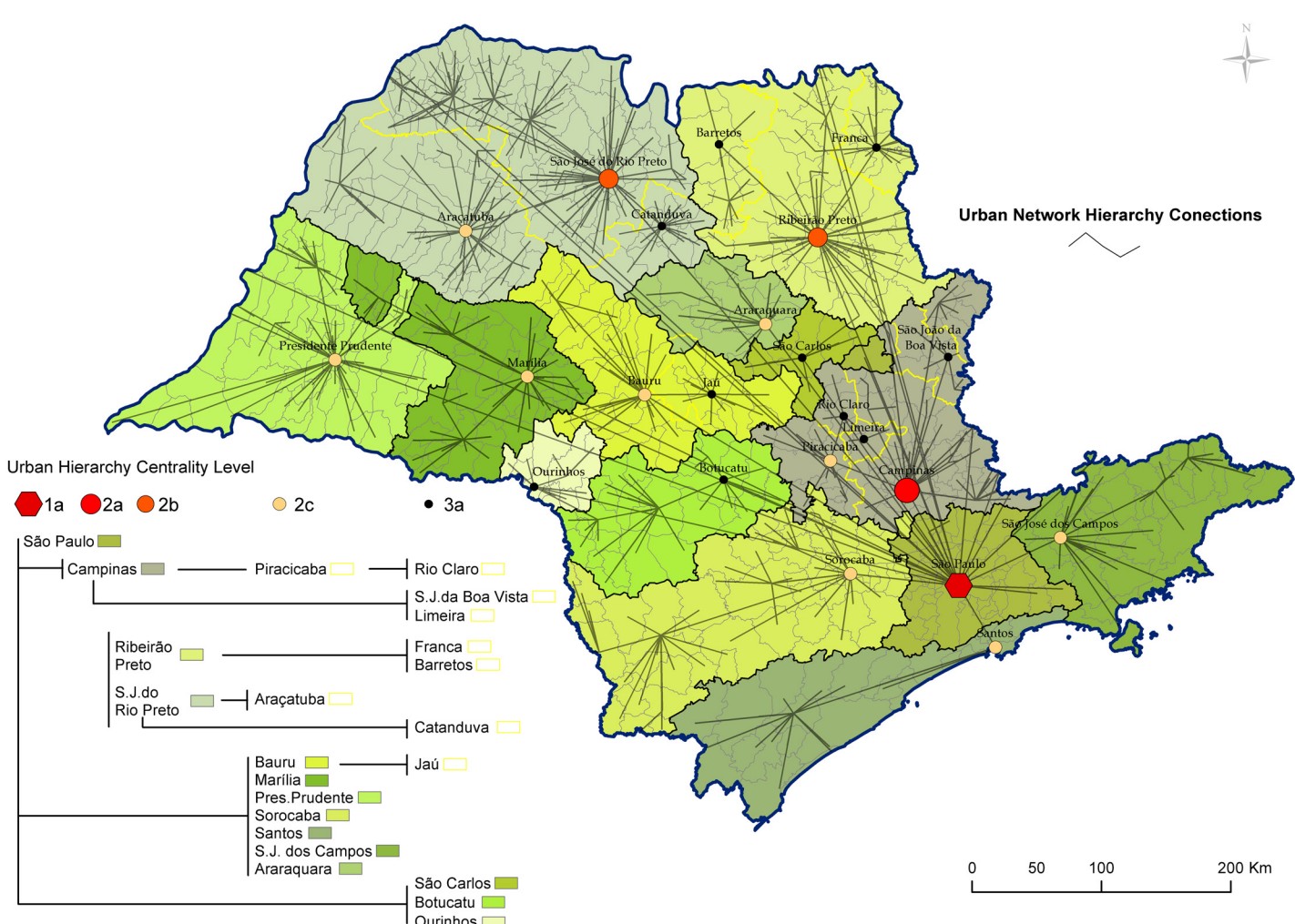

**Fig 3. Level of urban hierarchy find at São Paulo State: Metropolis, regional capital, subregional center, zone centers, and local center.** The tree diagram schematizes the hierarchy of the main cities, and the lines that divided the territory (the yellow and black borders) show the regions of influence of cities classified as level 1, 2, and 3 of urban hierarchy. The map was made using the software ArcGIS (version 10.8).

the study period; therefore, it is a first insight into disease dispersion. The white areas on the map show the municipalities without COVID-19 cases, and at the same time, far away from the ones where the disease was already reported. Overlapping this map with the one shown in Fig 1, we can see that most municipalities without case reported are classified as rural ones. The main roads that cross the state are also highlighted, and we hypothesize is that they also play an important role in the disease spreading over the territory.

To understand the regional pattern of SARS-CoV-2 spreading Fig 5, presents the elementary spatial structures identified as the main ones responsible for the disease spread inside the state. They comprise the main roadways as well as the airports that give people and trade mobility, and the hotspots of the disease introduction and spread. The airports and the cities are displayed by circles proportional to their role on COVID-19 spread. In the case of cities, we classified them as principal (São Paulo, Campinas, São José dos Campos, Ribeirão Preto, and São José do Rio Preto) and secondary (Santos, Araçatuba, Presidente Prudente, Bauru, Marília, São Carlos, Sorocaba, Rio Claro, and Piracicaba) urban centers in the level of relevance for the disease spread. Cities belonging to the metropolitan area, such as Santos, São

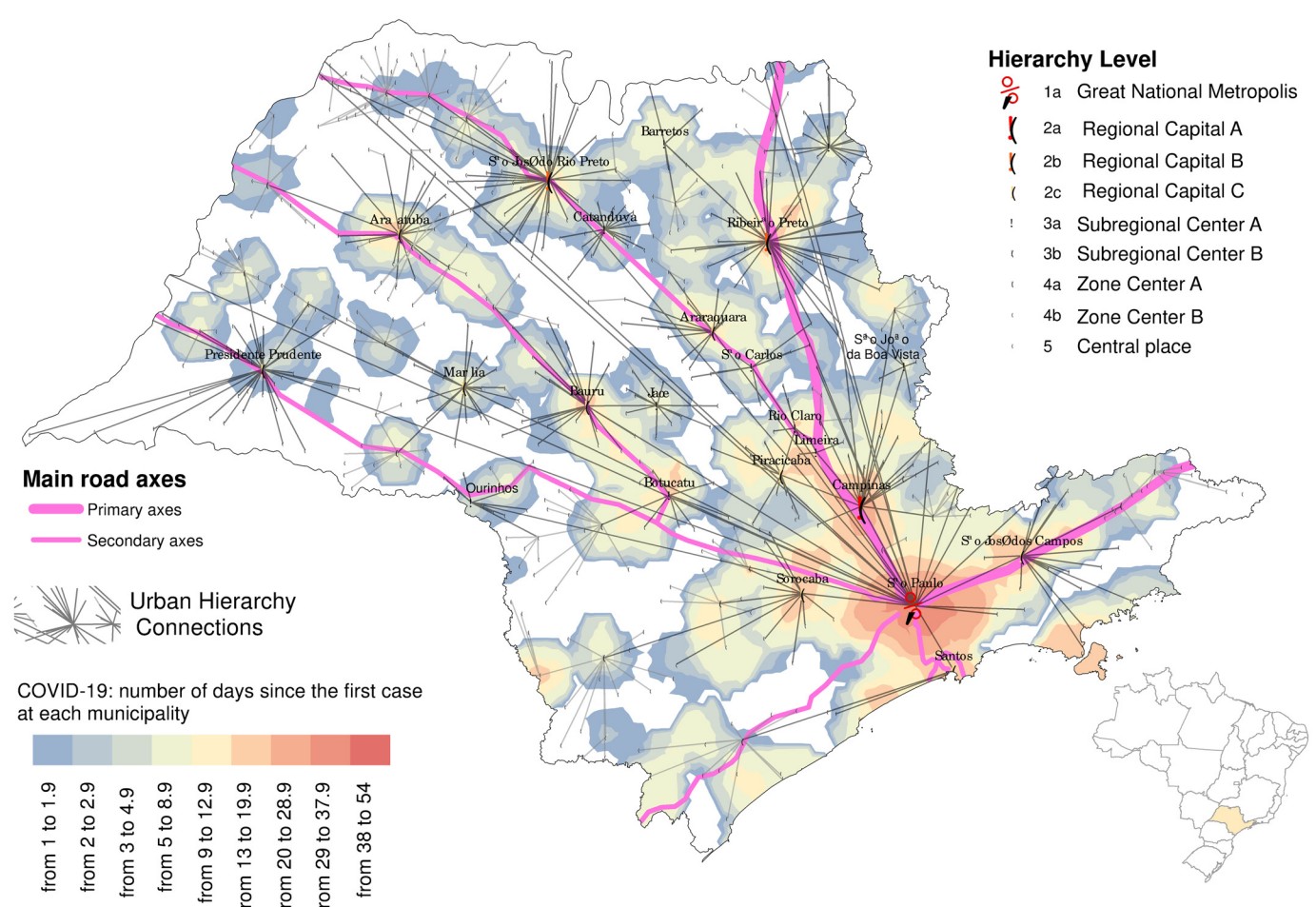

**Fig 4. Dispersion map for COVID-19 in São Paulo State Brazil from March 25 to April 18, 2020.** The color spectrum indicates early introduction areas (in red) to those of the more recent COVID-19 introduction (in blue). The main roadways that cross São Paulo State and the urban hierarchy level of each municipality is displayed. The map was made using the software ArcGIS (version 10.8).

José dos Campos, and Campinas, are classified as "contiguity"; the other ones are connected to São Paulo City through a primary or secondary axis. The metropolitan area of São Paulo and the state's region where there is a massive concentration of elderly population (older than 60 years of age) are highlighted. The latter is called vulnerable because disease lethality among them is high. For these listed cities, demographic characteristics, number of reported cases and, disease lethality (up to April 18, 2020) are presented in Table 1. Santos, which has a considerable mortality per 100,000 inhabitants, is the one in the list with the more significant number of the older population ($\geq$ 50 years).

Based on the results of the exploratory analysis (Figs 1 and 4) and population mobility studies (Figs 2 and 3), two dispersion patterns were postulated. In the first one, virus dispersion occurs by contiguity, from a region of initial introduction, that is the Metropolitan Region of the Capital, the City of São Paulo (contagious diffusion) to its nearest neighborhoods. In the second one, there is a long-distance dispersion following structural axes (roadways and airways) that connect São Paulo city to peripheral municipalities of regional importance (hierarchical diffusion). From these, diffusion by contiguity occurs again to smaller municipalities.

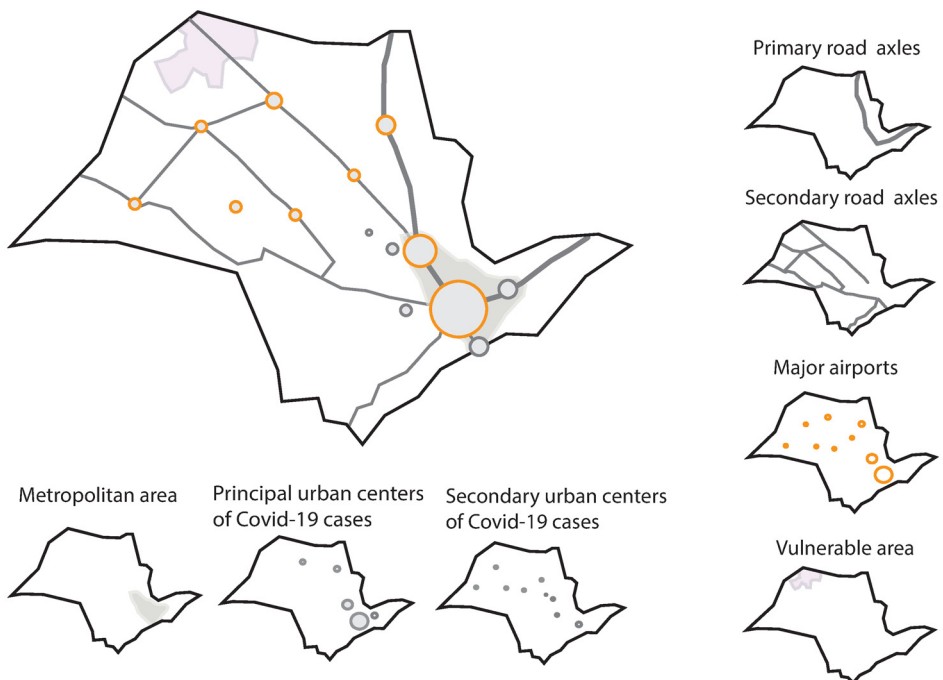

**Fig 5. Elementary spatial structures associated with COVID-19 spread in São Paulo State, Brazil.** Around the main map, we display the structures that comprise it, such as road axes, regional airports, the metropolitan area of São Paulo city, municipalities keys as centers of disease dispersion from the metropolitan area to inner state, and municipalities where the oldest population of São Paulo lives.

**Table 1. Epidemiologic COVID-19 data for São Paulo State capital and hotspots cities for disease introduction and spread on April 18th (see Fig 5).**

| Municipality | Population (inhabitant) | Dist. [1] (Km) | Connection with the capital[2] | Cumul. cases | Incid.[3] | Cumul. deaths | Mort.[3] | Date of arrival[4] |
|---|---|---|---|---|---|---|---|---|
| São Paulo (capital) | 12252023 | - | - | 9428 | 76.95 | 686 | 5.6 | 2020-03-25 |
| Campinas | 1204073 | 95 | Contiguity | 184 | 15.28 | 7 | 0.58 | 2020-03-18 |
| Ribeirão Preto | 703293 | 314 | Primary axis | 76 | 10.81 | 5 | 0.71 | 2020-03-26 |
| São José do Rio Preto | 408558 | 440 | Seconday axis | 70 | 17.13 | 4 | 0.98 | 2020-03-18 |
| São J. dos Campos | 721944 | 91 | Contiguity | 138 | 19.12 | 3 | 0.42 | 2020-03-18 |
| Santos | 433311 | 55 | Contiguity | 287 | 66.23 | 19 | 4.38 | 2020-03-30 |
| Sorocaba | 679378 | 100 | Secondary axis | 41 | 6.03 | 8 | 1.18 | 2020-03-27 |
| Piracicaba | 404142 | 162 | Secondary axis | 19 | 4.70 | 2 | 0.49 | 2020-03-30 |
| Bauru | 376818 | 343 | Secondary axis | 50 | 13.27 | 3 | 0.80 | 2020-04-03 |
| Presidente Prudente | 228743 | 550 | Secondary axis | 6 | 2.62 | 2 | 0.87 | 2020-04-08 |
| Araçatuba | 197016 | 530 | Secondary axis | 37 | 18.78 | 0 | 0.00 | 2020-03-31 |
| Marília | 216745 | 438 | Secondary axis | 8 | 3.69 | 1 | 0.46 | 2020-04-03 |
| São Carlos | 251983 | 231 | Primary axis | 7 | 2.78 | 2 | 0.79 | 2020-04-06 |
| Rio Claro | 186273 | 176 | Secondary axis | 14 | 7.52 | 3 | 1.61 | 2020-04-03 |

[1]. distance from the capital;

[2]. classification according to Fig 5;

[3]. incidence or mortality per 100,000 inhabitants.

[4]. Date of the disease arrival at each municipality.

A relationship between disease spreading and territorial geography was also established in other epidemics [22, 25]. Differently from São Paulo State [26], showed that the first wave of SARS-CoV-2 pandemic in Germany followed a dispersion pattern called relocation diffusion process since the arrival of infections in Germany coincided with a traditional carnival festivity. Therefore, a single infected individual transmitted the infection to several others. After the festivities, people went back to their homeland, creating long-range connections, and new spots of infection spread, which were randomly distributed across the country. In São Paulo State, since all non-essential activity was limited, the spread followed the main routes of commercial relationships and supply distribution, in a hierarchical diffusion, firstly reaching the most important cities in São Paulo State, and locally spreading within their regions.

Currently, SARS-CoV-2 is reported in all São Paulo territorial; the last city to be achieved by this coronavirus (September 1th) was Santa Mercedes, a rural municipality with 2,945 inhabitants and 580 km far away São Paulo City (also out of the main roads of coronavirus dispersion). On September 27th, São Paulo State reports 972,237 confirmed cases and 35,108 deaths. The isolation index is 48%, the Intensive Care Unit (ICU) occupation is 45,6%, and disease lethality is 3,6% [27]. Schools and universities still closed, and the state has its own plan of quarantine measures ("Plano São Paulo") that, based on the growth rate of COVID-19 cases and deaths and bed occupancy rates in each Regional Health Departments (DRS), can be more or less flexible. Cities belonging to the same DRS (we have seventeen) are ruled by the same quarantine measures. Phase 1 is considered a contamination phase, and only essential services are permitted. Phase 2 is considered an attention phase with the possibility of some services such as commerce opening. Food courts are still banned in this phase. Phase 3 is considered a controlled phase with some flexibilization. Phase 4 has less restriction than phase 3, and at phase 5, all services are allowed to open, maintaining all specific protocols. Now, in December 2020, all São Paulo State is at phase 3 [28].

Our predictions of routes and risks of COVID-19 in inner São Paulo State (Fig 5) have been thus far validated by surveillance data. Given the extensive mobility between smaller municipalities and those cities with regional economic relevance [17], it is reasonable to infer that the regional spread of SARS-CoV-2 infections depends on the success of non-pharmacological strategies applied in the latter. We also state that similar methodological approaches can direct public health strategies in other developing countries, especially those that either have great territorial extension and/or have diverse patterns of urbanization and mobility.

Limitations of the analysis include: (i) the no-identification of asymptomatic individuals and, potentially, mild or moderate infectious, since only symptomatic cases that seek for medical assistance have been tested; (ii) data dependence, *i.e.* data set does not distinguish between imported and autochthonous cases; (iii) the assumption that all individuals have the same degree of susceptibility and transmibility of the disease, regardless the environment they live; (iv) the transmission is homogeneous within the cities; (v) mitigation strategies are the same everywhere. All those characteristics may variate according to the city because the number of tests that is distributed and performed among cities is not homogeneous; the number of contacts among people changes according to the city characteristics, such as the use of public transportation [29]; and people's adherence to social distancing really differed across the state, which may be related to the epidemics delay into reach the small inner cities, affecting people's risk perception [30].

Moreover, the data source in Brazil has been updated with some delay, regarding the occurrence of the infections [31]. Nevertheless, since the data used in this study is related to the arrival of infections in each city, which happened in early 2020, we expect the numbers to be trustful at the point of the analysis. Despite there is no data at a granular level, such as

information about the address of the infection occurrence, the data is enough to perform the analysis and reach our goal, which was to study the spread of SARS-CoV-2 among cities.

## Conclusion

Spatial analysis of coronavirus spread is an important tool for public health management, as it can highlighting the main routes of disease dispersion and the fragility of municipalities related to its socio-demographic characteristics. In the case of São Paulo State, this analysis evidenced the hotspots and main routes of disease dispersion from capital to inner state. Currently, non-pharmacological controls are the only tools to halt or diminish the disease spreading among both individuals and municipalities. The existence of two different ways of disease dispersal, by standard diffusion and hierarchical one, can provide alternative strategies to control disease spread in the São Paulo territory.

This work shows that it was possible to understand and even predict the route of COVID-19 spread in São Paulo State looking to the cities' hierarchy, which means that the spread of the epidemic does not follow a diffusion process but reaches the cities based on their regional importance and activities. After that, the epidemic spread to contiguous cities following a diffusion standard process. We state that those cities are responsible for the arrival of the epidemics in the inner São Paulo State and demand attention.

## Supporting information

**S1 Data set.**
(XLSX)

## Author Contributions

**Conceptualization:** Carlos Magno Castelo Branco Fortaleza, Raul Borges Guimarães, Cláudia Pio Ferreira.

**Data curation:** Carlos Magno Castelo Branco Fortaleza, Raul Borges Guimarães, Rafael de Castro Catão, Cláudia Pio Ferreira, Gabriel Berg de Almeida, Thomas Nogueira Vilches, Edmur Pugliesi.

**Formal analysis:** Carlos Magno Castelo Branco Fortaleza, Raul Borges Guimarães, Rafael de Castro Catão, Cláudia Pio Ferreira, Gabriel Berg de Almeida, Thomas Nogueira Vilches, Edmur Pugliesi.

**Funding acquisition:** Cláudia Pio Ferreira.

**Investigation:** Carlos Magno Castelo Branco Fortaleza, Raul Borges Guimarães, Rafael de Castro Catão, Cláudia Pio Ferreira, Edmur Pugliesi.

**Methodology:** Carlos Magno Castelo Branco Fortaleza, Raul Borges Guimarães, Rafael de Castro Catão, Cláudia Pio Ferreira, Gabriel Berg de Almeida, Edmur Pugliesi.

**Project administration:** Carlos Magno Castelo Branco Fortaleza, Raul Borges Guimarães.

**Software:** Edmur Pugliesi.

**Supervision:** Carlos Magno Castelo Branco Fortaleza, Raul Borges Guimarães, Cláudia Pio Ferreira.

**Validation:** Carlos Magno Castelo Branco Fortaleza, Raul Borges Guimarães, Rafael de Castro Catão, Cláudia Pio Ferreira, Gabriel Berg de Almeida, Edmur Pugliesi.

**Visualization:** Carlos Magno Castelo Branco Fortaleza, Raul Borges Guimarães, Rafael de Castro Catão, Cláudia Pio Ferreira, Gabriel Berg de Almeida, Edmur Pugliesi.

**Writing – original draft:** Carlos Magno Castelo Branco Fortaleza, Raul Borges Guimarães, Cláudia Pio Ferreira, Gabriel Berg de Almeida.

**Writing – review & editing:** Carlos Magno Castelo Branco Fortaleza, Raul Borges Guimarães, Rafael de Castro Catão, Cláudia Pio Ferreira, Gabriel Berg de Almeida, Thomas Nogueira Vilches, Edmur Pugliesi.

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
