## [Decision Letter · Decision Letter 0]

24 Jul 2020

PONE-D-20-12111

Elementary spatial structures and dispersion of COVID-19: health geography directing responses to public health emergency in S~ao Paulo State, Brazil

PLOS ONE

Dear Dr. Fortaleza,

Thank you for submitting your manuscript to PLOS ONE. After careful consideration, we feel that it has merit but does not fully meet PLOS ONE’s publication criteria as it currently stands. Therefore, we invite you to submit a revised version of the manuscript that addresses the points raised during the review process.

We look forward to receiving your revised manuscript.

Kind regards,

Javier Sanchez

Academic Editor

PLOS ONE

Additional Editor Comments:

The comments from two reviewers indicated that the current publication needs a major revision. One of the reviewers recommended the rejection of the manuscript but provided a detail report indicating the major concerns with this manuscript. In addition to the comments provided by the two reviewers, the authors should provide a clear description of the SEIR model, a solid justification of the parameters used and a validation of that model. When addressing the reviewers comments, please keep in mind that one of the major flaws of this current version of the manuscript is related to criterion #3 of the publication list criteria ("Experiments, statistics, and other analyses are performed to a high technical standard and are described in sufficient detail. Experiments must have been conducted rigorously, with appropriate controls and replication. Sample sizes must be large enough to produce robust results, where applicable. Methods and reagents must be described in sufficient detail for another researcher to reproduce the experiments described.") Therefore, in the revised version this part needs be improved significantly in order to accept this work for publication.

Journal Requirements:

2. We noted in your submission details that a portion of your manuscript may have been presented or published elsewhere.

"This manuscript is being submitted in the present form to medXriv"

Please clarify whether this publication was peer-reviewed and formally published. If this work was previously peer-reviewed and published, in the cover letter please provide the reason that this work does not constitute dual publication and should be included in the current manuscript.

3. We note that Figure 1,2,3 in your submission contain map images which may be copyrighted. All PLOS content is published under the Creative Commons Attribution License (CC BY 4.0), which means that the manuscript, images, and Supporting Information files will be freely available online, and any third party is permitted to access, download, copy, distribute, and use these materials in any way, even commercially, with proper attribution. For these reasons, we cannot publish previously copyrighted maps or satellite images created using proprietary data, such as Google software (Google Maps, Street View, and Earth). For more information, see our copyright guidelines: http://journals.plos.org/plosone/s/licenses-and-copyright.

3.1.    You may seek permission from the original copyright holder of Figure 1,2,3 to publish the content specifically under the CC BY 4.0 license.

3.2.    If you are unable to obtain permission from the original copyright holder to publish these figures under the CC BY 4.0 license or if the copyright holder’s requirements are incompatible with the CC BY 4.0 license, please either i) remove the figure or ii) supply a replacement figure that complies with the CC BY 4.0 license. Please check copyright information on all replacement figures and update the figure caption with source information. If applicable, please specify in the figure caption text when a figure is similar but not identical to the original image and is therefore for illustrative purposes only.

Reviewers' comments:

Reviewer's Responses to Questions

**Comments to the Author**

1. Is the manuscript technically sound, and do the data support the conclusions?

Reviewer #1: No

Reviewer #2: Yes

2. Has the statistical analysis been performed appropriately and rigorously? 

Reviewer #1: No

Reviewer #2: N/A

3. Have the authors made all data underlying the findings in their manuscript fully available?

Reviewer #1: Yes

Reviewer #2: Yes

4. Is the manuscript presented in an intelligible fashion and written in standard English?

Reviewer #1: Yes

Reviewer #2: Yes

5. Review Comments to the Author

Reviewer #1: Summary of the research and overall impression:

This is an ecological study that uses a geographical model of population mobility to explain the pattern of spread of SARS-Cov-2 infection in the state of São Paulo, Brazil.

The study used surveillance data to identify two patterns of spread: one by contiguity and the other by highways from the capital to the inner state.

The article presents only one conclusion: the main route of dispersion from the capital to the interior is through highways. Considering that in the state of São Paulo the movement of people between the cities is done almost exclusively by highways there would be no other possibility of dispersion. Therefore, there is no new or relevant information produced by the article. In addition, the methodological problems described below make me recommend rejecting the article.

aspects to improve:

Line 30: It is important to have a description of the quality of the database and the sensibility of the surveillance system. It must be made clear what the extent of any underreporting and its impact on the results obtained.

It is necessary to describe the sample processing capacity, the population tested and possible spatial heterogeneity in the capacity to perform diagnostic tests.

Do municipalities have the same capacity to perform tests? What is the impact of these factors on the results?

It is known that in Brazil there is a significant delay in confirming cases so it is necessary to clarify which date was used in the analysis: the date of onset of symptoms or confirmation of diagnosis?

Lines 42-45: Regarding the metholodogy to create the diffusion map (figure 1), it assumes that once the first case is detected in a municipality, there would be cases around it, defined by the interpolation method used.

Is this epidemiologically reasonable?

This initial case could be a person coming from an infected municipality far away having no relation with the infected neighbouring municipalities. In this case, what would be the epidemiological meaning of this interpolation?

In addition, there is no justification for choosing 6 nearest neighbors with reported cases for interpolation. Different choices would generate different patterns. What is the justification for this choice (6 municipalities)?

This choice may determine, in some situations, the selection of very distant municipalities and possibly without any epidemiological link with the municipality of the case. In this situation, what would be the epidemiological meaning of this interpolation?

What is the advantage of this methodology when compared to the traditional construction of heat maps at pre-defined time intervals to characterize in space and time the evolution of the disease?

Lines 57-59: "In the second step, data about each municipalities such as infrastructure, facilities, land use, jobs, and urban mobility were used to identify the fundamental entities of the spatial structure that triggers coronavirus dispersion". These analyses were not presented.

Lines 95-96: the authors postulate two mechanisms for dispersion of the disease based on the result obtained. However, are not these mechanisms widely known and often identified in epidemics? Why would this represent new information?

Lines 102-103: Figure 2 presents diffusion axes classified as primary and secondary without any methodology being mentioned to justify this classification. The same occurs with municipalities that are classified as major centers of spatial diffusion and secondary centers of spatial diffusion. Is this classification based simply on the number of cases reported or has the potential for dissemination of these municipalities been assessed in any way?

Lines 104-107 and Fig 4 and 5: Since the parameters used in the simulations are the same, with the exception of population size, it is expected that the dynamics will be the same, except for a scale factor. Wouldn't the use of just one graph be enough to represent the flattening of the curve?

Lines 119-120: the authors say "our prediction of routes and risks of COVID-19 in inner São Paulo State (Fig 2) have been thus far validated by surveillance data (Fig 3)". However, considering that the model was generated from the surveillance data, it cannot be considered that there has been any validation here. The model only recovers the initial information used.

Figure 3: there is a series of mapped information whose origin is not explained in the text. What is the meaning of strongly connected urban municipality, secondary connected urban municipality and rural municipality and how this information relates to the article. The axes of virus dispersion also did not have their estimation methodology described in the text.

Lines 135-136: As in the state of São Paulo, people move towards the countryside almost exclusively by road since there is no significant transport of people by plane, train or waterway, what other possibilities would exist besides roads?

Reviewer #2: Overall

The study is using spatial analysis and compartmental modeling of populations to determine dispersion of COVID-19 in Sao Paulo, Brazil. While applicable methods were used and the work has significant merits, the manuscript needs improvements to be eligible for a publication. The improvements include adding detailed descriptions of methods, assumptions, and reasoned interpretations.

The key points are:

a. Spatial analysis was used to illustrate the spread of COVID-19 in Sao Paulo over time and over major locomotion routes

b. SEIR compartmental model was used to model epidemic curves for the 645 municipalities. The population was stratified into 15 different age groups. The results were summarized by the 18 mesoregions of Sao Paulo

c. Two scenarios were modeled: with social distancing and without social distancing

d. The R0 for scenario 1 (without social distancing) was 2.7 and the researchers have assumed 50% reduction of the contact rates for the scenario with social distancing

e. The authors interpret the results by generally recommending social distancing in peripheral municipalities to reduce the spread of COVID-19

Title is not explanatory of the analysis. It could be shortened and made precise. Suggestion “The use of health geography and compartmental modeling to understand early dispersion of COVID-19 in Sao Paulo, Brazil”

Abstract

1. The hypothesis and objective aren’t clear. How does understanding the spatial patterns of dispersion from the urban metropolitan area benefits understanding the dispersal in non-metropolitan inner municipalities?

2. If the hypothesis is that disease spread occurred along the major routes of locomotion from the capital and metropolitan area to the periphery, mention this clearly

3. While the recommendation for social distancing and its impact in reducing the disease is widely known, please discuss how this specific analysis translates to inform disease control?

Introduction

Please describe the hypothesis and objectives. While the merit of the work is recognizable, objectives and hypothesis aren’t clear.

Methods

1. Line 46: Add “influence on the predicted value than those far away..”

2. Line 50: Type “April” and the dates described in Abstract and data are different (15th vs 18th of April) given the exponential nature of cases and hospitalizations observed with COVID-19, confirming the correct date would matter

3. Line 57-59: Please describe how these features were used in the analysis.

4. Line 62: Mention how the SEIR results for 645 municipalities were summarized by the mesoregions it seems (n=18)

5. Line 67: mention that its “..fifteen age groups” and please justify this extensive stratification of age groups with a reference

6. Provide relevant reasoning for the initial 10 cases for the age categories 25 – 50 age class

7. Line 88-be more specific what "disease control" entails-does this differ from "social distancing" references in line 75?

8. Line 108: Heterogeneity of what characteristic? Age categories?

9. SEIR model assumptions and definitions are not mentioned explicitly

a. Provide a reference for the choice of 2.7 as R0

b. Are you assuming the rates of transmission in the inner municipalities is comparable to capital and the metropolitan areas? If yes, please explain why this assumption was made

c. Do you assume same parameter values for all age groups? Please mention if this assumption was made. Except for Lines 79-82 most other assumptions seems same for all age groups

10. It is unclear how the contiguity, primary and secondary axis were defined when interpreting SEIR model results (Table 2 and Fig 3). Include details on the definitions and please describe.

11. The definitions of axis of dispersion along the major locomotion routes may involve assigning a relative time connection matrix to recognize the average direction of the disease spread over time.

12. What software/s were used to perform the analysis and illustrate?

Results & Discussion

1. Line 92-94: Please revise the sentence. Typo: “de”

2. Line 100: “peripheral” municipalities in lieu of ‘pole’ municipalities

3. Explain the details depicted in Fig 2 in detail in the text.

4. Please explain the key limitations related to the data. The reference 11 is the database it seems and it is unclear to the reader what are the limitations

5. Lines 125 – 126: Please explain how does the pressure from industry and trading companies are affecting the social distancing requirements

6. While the recommendation for social distancing and its impact in reducing the disease is widely known, please discuss how this specific analysis translates to inform disease control? What distance from the capital/metropolitan area got highly affected, within what timeframe, and the major two routes of locomotion identified through the analysis as mainly involved in the dispersal, does these routes have specific characteristics?

 

Tables and figures

Fig 1.

- Font size of the legend need to be increased.

- Include a scale bar and garticules

Fig 2.

- The figure is too busy with multiple sub figures and no clear legend explaining the map details. For example what does the size of circular symbols represent?

- Include a scale bar for the main figure

- “Secondary” typos in two places

Fig 3.

- While Table 2 description claims to have defined the variable ‘Connection with the Capital’ based on Fig 3. The figure does not illustrate what areas are considered as ‘Contiguity’, ‘Primary Axis’ and ‘Secondary axis’. Please clarify and change the figure and table labelling

- Define what do you refer to as ‘axis’ in the text. The figure shows two different information and both of these are labelled as ‘axis’

o ‘Axes of virus dispersion’: Standard deviation ellipses has major and minor semi axis and a rotation. Please mention these numerically in a table.

o “Main routes of COVID-19 dispersion’ . This sounds synonymous to the previous, except that the features are recognizing the major roads that contribute to the disease spread. It is unclear what does the thickness of the lines represent. Describe and relabel accordingly.

Fig 4 and 5.

- Please present them as one figure

- If there were 18 mesoregions in the analysis, as seen in Table 2, why does the graphs show only 17? Why does the figures exclude ‘Registro’?

Table 2.

- Column 4: remove the capitalization of ‘Capital’ as it is not consistent with the other column names

Supplementary data

1. Present the data column names in English

2. Include a metadata sheet explaining the data

a. column names

b. the age groups

c. color codes

d. the case numbers

6. PLOS authors have the option to publish the peer review history of their article (what does this mean?). If published, this will include your full peer review and any attached files.

Reviewer #1: **Yes: **FERNANDO FERREIRA

Reviewer #2: No

---

## [Author Response · Author response to Decision Letter 0]

11 Nov 2020

We thank the reviewer’s comments. We recognize that they helped to improve the paper.

Reviewer #1: 

The first four questions are more or less correlate, we answer it here one by one but in the text we put all the information together in one paragraph (lines 54-76).

1) Line 30: It is important to have a description of the quality of the database and the sensibility of the surveillance system. It must be made clear what the extent of any underreporting and its impact on the results obtained.

We added the following sentence in the text to clarify it (lines 54-67): “The data was obtained from Brasil IO’s compiled databank (https://brasil.io/dataset/covid19/boletim/) that is kept by a task force of volunteers (researchers and reporters). This group daily catches, from the epidemiological reports of each state, the number of confirmed cases and death by SARS-CoV-2 and let it publicly available. Because of the recognition of SARS-CoV-2 as a pandemic by WHO, laboratory certification in Brazil ranged from few laboratories to 26, in eight weeks since the beginning of the epidemic; the majority is located at São Paulo State (Grotto, R., et al, 2020. Increasing molecular diagnostic capacity and COVID-19 incidence in Brazil. Epidemiology and Infection,148, E178. doi:10.1017/S0950268820001818). Data reporting of Severe Acute Respiratory Illness (SARI) is mandatory in Brazil. A specific form (national database SIVEP-Gripe) collects information that allows us to estimate reported delay, disease fatality at which age class, and identify confirmed cases of the disease. This permits surveillance of all respiratory diseases in Brazil. Only cases that were hospitalized belong to this data set, therefore, underreporting is expected. On the other hand, this is probably homogeneous along the municipalities and hence, will not impact on the observed pattern of disease spreading.”. To detect underreporting, a broader serological survey can be done. We have one in course in Brazil which is in its fourth round, but for the propose of this study this data cannot be used, since they are recent. 

2) It is necessary to describe the sample processing capacity, the population tested and possible spatial heterogeneity in the capacity to perform diagnostic tests.

We added the following sentence in the text to clarify it (lines 72-76): “Lastly, test capacity has grown fastly in São Paulo State; currently 13 out of 35 laboratories of Brazil are settled in this state. To detect underreporting and follow the population susceptibility along the course of epidemic, a broader serological survey is on course in Brazil. This may help in defining target groups to vaccination.” Spatial heterogeneity related to the capacity of testing can be a problem when we look for all the country; at São Paulo state the epidemic took time to move from capital to the countryside because of the restriction done at the beginning of the epidemic. This gave us time to improve test capacity and hospital-care structure in the municipalities far away from the metropolitan region. 

3) Do municipalities have the same capacity to perform tests? What is the impact of these factors on the results?

We added the following sentence in the text to clarify it (lines 68-72) “Out of this national surveillance system (SARI), test capacity can varies among the cities because many of them make agreements with factories to test the population of the site where the factory is settled. The same procedure has been done in universities, schools, and firms that returned their activities. But this data is not considered here, since SARI only reports severe cases.” 

4) It is known that in Brazil there is a significant delay in confirming cases so it is necessary to clarify which date was used in the analysis: the date of onset of symptoms or confirmation of diagnosis?

The data used in the analysis is related to the confirmation of diagnosis, which, unfortunately, is not the best one to be used. The best one is the data of onset of symptoms with delay corrected by nowcasting procedures. But until now, it is hard to have access to data of COVID-19 in Brazil. At lines 53-54 we specify it “we used surveillance data (number of confirmed cases of coronavirus) updated on April 18th, 2020”. 

5) Lines 42-45: Regarding the methodology to create the diffusion map (figure 4), it assumes that once the first case is detected in a municipality, there would be cases around it, defined by the interpolation method used. Is this epidemiologically reasonable?

Based on the comments, we changed the methodology (please check lines 140-167). In particular, we highlight (lines 137-138): “The fourth map was constructed by interpolating over the total number of days at which coronavirus transmission was reported in each municipality”, and (lines 155-165): “Although we are using the time-lapse since the first case reported in each municipality to create the interpolated surface that permits us to predict the epidemic course along the state, some cities entered community transmission only after 10 to 15 days from the first confirmed case. This reflects not only the stochastic nature of the process of introduction of a new pathogen in a community, but also the fact that data of COVID-19 in Brazil do not distinguish between imported and autochthonous cases. This is an exploratory analysis that permits us to follow the dispersion pattern of the virus, glimpsing the next cities that will probably be affected by the disease. The palette of colours of the map comprises the period from February 25th to April 18th, with white color indicating disease absence.”

6) This initial case could be a person coming from an infected municipality far away having no relation with the infected neighboring municipalities. In this case, what would be the epidemiological meaning of this interpolation?

We change the way at which the interpolation was done and the parameters related to the procedure of interpolation, in order to visualize closer influences. In any case, the exact and deterministic method do not predict when the next cases will be, but instead, organizes the main structure of disease dispersion showing the contiguity of the confirmed places with the additional layer of municipalities with and without cases. At that moment (April 18th, 7th week from the first confirmed case in Brazil) the confirmed cases were already reported in 227 municipalities of São Paulo (35% of total), mainly in areas close to the capital and regional centres. Besides, the epidemiological link in this case is not especially important as we were seeking for geographical links, as the proximity to major cities (transport hubs) and cities hierarchy and its influence over the other ones, as an attempt of understand how space is important to the spread of the disease. The interpolation had more connection to understand a regional pattern that, with the other two maps, indicated the spatial structures (such as airports, roads, demographic density, ratio between rural and urban population) that conditionate this observed pattern. We also added the following sentence in the text to clarify it (lines 171-173): “we want to emphasize that we are not looking for epidemiological links that explain disease transmission among municipalities, but we are seeking geographical links that conditionate the regional pattern of disease spreading along São Paulo territorial”.

7) In addition, there is no justification for choosing 6 nearest neighbors with reported cases for interpolation. Different choices would generate different patterns. What is the justification for this choice (6 municipalities)?

The diffusion map was planned to be a simple deterministic construction, built to represent the regional pattern of the first confirmed records of COVID-19. The neighbors were select by a parsimonious method, after an exploration of data. Now, we corrected it by searching the literature for a quantitative method that permitted us to optimize the neighborhood in order to create a representative pattern. We redone the analysis using the root means squared error to compare among different number of neighbors, and shapes of neighborhood. This analysis optimizes the size of the neighborhood, and the new figure is plotted with minimum and maximum values of 2 and 12 neighbors searched in a circle of 25 km. We rewrite the description of the method to clarify it (lines 140-148). Observe that we are not making any more the interpolation over the date of the disease arrival in each city but over lapse since the arrival. 

8) This choice may determine, in some situations, the selection of very distant municipalities and possibly without any epidemiological link with the municipality of the case. In this situation, what would be the epidemiological meaning of this interpolation?

As comment before, the analysis was done to understand the geographical links that drive the spatial-temporal pattern of COVID-19 at São Paulo State, highlighting the regional importance of each municipality, and driving government resources to epidemic control. We chose the circle as the shape of the neighborhood for the interpolation procedure. The radius of the circle, the minimum and maximum numbers of neighbors, were respectively 25 km, 2 and 12. Besides the fact that the number of neighbors can vary, the contribution of each neighbors was weighted by the inverse of its distance. We added the following sentence to explain it (lines 253-256): “The colors have to be reading such as a predictor of an earlier or later arrival of the disease in each city of the map because they comprise only the period of study; therefore, it is a first insight into disease dispersion”

9) What is the advantage of this methodology when compared to the traditional construction of heat maps at pre-defined time intervals to characterize in space and time the evolution of the disease?

The objective of the two approaches are different. Heatmaps create a spatial-colored map at a fixed time where the different colors are associated with the amount of data you have on a specific spatial per-defined range (density function visualization). Using this approach, to understand the spatial diffusion pattern of COVID-19 at São Paulo state, we need to compare several heat-maps constructed at different time scales; therefore, a time scale must be defined. In summary, we lose the information related to the time scale, and the power of prediction it. Using Inverse distance weighting (IDW) technique we can visualize the spatial distribution in one single map. 

10) Lines 57-59: "In the second step, data about each municipalities such as infrastructure, facilities, land use, jobs, and urban mobility were used to identify the fundamental entities of the spatial structure that triggers coronavirus dispersion". These analyses were not presented.

To solve this problem, we added the following paragraph to the text, and new references and Figure 2. We want to highlight that the information can be get from IBGE site (census data done by the federal government and thematic studies) organized in shapes that can be downloaded. Part of the information is in English, but the large amount of it is in Portuguese. Please see (lines 81-101): “This information was compiled from census data done by the Federal Government and other thematic studies. The metropolis of São Paulo appears as the largest urban complex in the country, with almost 22 million inhabitants and a high level of integration with other municipalities that comprise the national territory. It is listed as an alpha global city by the Globalization and World Cities Research Network (GaWC). In the second level of the hierarchy, we have Rio de Janeiro and Brasília (the capital of Brazil). Focusing on São Paulo State, it has two main axes of urban and trade mobility (roadways, railways, and airways), the first one connecting São Paulo with Rio de Janeiro, and the second one connecting São Paulo with Brasília and Central-West Region of Brazil. Besides, these main transportation axes, we have a secondary flux network connecting the metropolis of São Paulo to country municipalities, and the South of Brazil. This secondary flux aggregates roadways, railways, airways, and waterways (Tietê-Paraná). Over this intense flux of people and trade, a complex structure of cities emerges, reinforcing this network composed of high hierarchy cities (as nodes) linked by the best transportation system of the country (as edges). With almost 48 million inhabitants, São Paulo State concentrates 23.6\\% of country's population and 33\\% of its income. Besides, São Paulo State has the highest number of primary (Metropolis) 2/15, and secondary (Regional Capital) 26/97 cities on the urban hierarchy level in Brazil, and high number of other cities classified as Subregional Center 77/352, and Zone Center 51/398, respectively, at tertiary and quaternary levels.” 

11) Lines 95-96: the authors postulate two mechanisms for dispersion of the disease based on the result obtained. However, are not these mechanisms widely known and often identified in epidemics? Why would this represent new information?

The central question was whether the spatial-temporal distribution of COVID-19 cases was due to hierarchical spread, contagious spread, or mixed diffusion. In the case of COVID-19, at São Paulo state, the epidemic follows a combined pattern, with a hierarchical diffusion (without contiguity, with leaps) in a first moment, and a contagious pattern in a second moment. In the case of direct-transmitted disease, a mixed diffusion is the common observed spatial-temporal pattern. Specifically for COVID-19, those two moments were distinguished (now it is not anymore) because of the travel restriction and quarantine imposed in the beginning of the epidemics. From the public police point of view understanding the diffusion and its velocity, helps on planning and management of the resources (hospitals and ventilators). This study was done in real-time, during the course of epidemic, and this represents information in loco to be used for decision taken. 

Besides, the epidemics have their own mechanisms of spreading that depends on a lot of factors, for example, if there is a vector involved on its transmission, how many different host exist and how important is each one on the disease transmission, what is the main path of pathogen entrance on the host, host susceptibility, how pathogen interact with the environment, and etcetera. This is a new pathogen, and it is not true that when know how it will behave, in fact we are learning to respond to it as epidemic evolves, and because of it a lot of decisions made at the beginning changed during epidemic course. 

12) Lines 102-103: Figure 4 presents diffusion axes classified as primary and secondary without any methodology being mentioned to justify this classification. The same occurs with municipalities that are classified as major centers of spatial diffusion and secondary centers of spatial diffusion. Is this classification based simply on the number of cases reported or has the potential for dissemination of these municipalities been assessed in any way?

The classification was done after crossing the number of cases with urban mobility resulting from healthcare (hospitals and clinics) and education (schools and universities) system use, economic activity, employment, and land use at each spatial localization. These are main routes of circulation, based on traffic and at beginning of epidemics most affected cities were near these roads.

The municipalities were classified with the centrality (assessed by IBGE for all urban network in Brazil) the major cities with higher hierarchies and that possessed cases were a primary, the lower hierarchies and presenting cases were classified as secondary. We added new figures and change the order of presenting them to clarity how this final map was constructed. 

13) Lines 104-107 and Fig 4 and 5: Since the parameters used in the simulations are the same, with the exception of population size, it is expected that the dynamics will be the same, except for a scale factor. Wouldn't the use of just one graph be enough to represent the flattening of the curve?

We agree that a unique graph would be enough to show the flattening of the curve. We redone them since now we have the parameters (such as mortality) in each municipality. Also, we changed the discussion approach to compare the simulations with the reported number of cases. Please check lines 306-314.

14) Lines 119-120: the authors say "our prediction of routes and risks of COVID-19 in inner São Paulo State (Fig 2) have been thus far validated by surveillance data (Fig 3)". However, considering that the model was generated from the surveillance data, it cannot be considered that there has been any validation here. The model only recovers the initial information used. 

We agree with the comment and rewrote the sentence. But we want to emphasize that now we are far away from the beginning of the epidemic in São Paulo State and the routes predicted by the study were validated with current data.

15) Figure 3: there is a series of mapped information whose origin is not explained in the text. What is the meaning of strongly connected urban municipality, secondary connected urban municipality and rural municipality and how this information relates to the article. The axes of virus dispersion also did not have their estimation methodology described in the text. 

We redone the figure and rewrite the its caption to clarify what we want to show. Also, we improved the description of it on the main text. We highlight in red the changes done in the manuscript, but also add here some points to be able to answer the reviewer comments. In particular, the axis of virus dispersion is given by the weighted standard deviation ellipses that account to directional influence. A common way of measuring the trend for a set of points or areas is to calculate the standard distance separately in the x and y directions. These two measures define the axes of an ellipse encompassing the distribution of features. The angle of rotation is measured taking into account the deviation of xy-coordinates from the mean center. The methodology is well know in statistics and we added a reference to it. Besides, ArcGIS has tools already constructed to measure it from the data. Please check lines 110-127. Also, data about each municipality such as territorial management, trade and services, financial services, health care services, educational institutions, media and communication markets, culture and sport, mode of transport, and land use were used to identify the fundamental entities of spatial structure that trigger coronavirus dispersion in São Paulo territory. These information were compiled from census data done by the Federal Government and other thematic studies. Please check lines 80-104.

16) Lines 135-136: As in the state of São Paulo, people move towards the countryside almost exclusively by road since there is no significant transport of people by plane, train or waterway, what other possibilities would exist besides roads?

São Paulo State concentrates an important flux of private/small airplanes coming from different States of Brazil besides its own airplanes flux (among São Paulo municipalities). Figure 2 was added to the manuscript to highlight it. This flux was not affected by the restriction mobility imposed in the country. Summarizing, besides roads, we have movement by many private planes. This is a characteristic of the São Paulo State that provides a huge number of industrialized and farm products for all the country, and also consume it. 

It is important to highlight that the dissemination of the virus was not a simple diffusion process but a hierarchical one. So, it is not a simple answer such as the disease will spread following the roadway’s direction, but also which cities will be affected first and when it will occur. 

Reviewer #2:

1) Title is not explanatory of the analysis. It could be shortened and made precise. Suggestion “The use of health geography and compartmental modeling to understand early dispersion of COVID-19 in Sao Paulo, Brazil”

Done.

2) Abstract

2A. The hypothesis and objective aren’t clear. How does understanding the spatial patterns of dispersion from the urban metropolitan area benefits understanding the dispersal in non-metropolitan inner municipalities?

2B. If the hypothesis is that disease spread occurred along the major routes of locomotion from the capital and metropolitan area to the periphery, mention this clearly

2C. While the recommendation for social distancing and its impact in reducing the disease is widely known, please discuss how this specific analysis translates to inform disease control?

We rewrote the abstract in order to take into account the comments: “Public health policies to contain the spread of COVID-19 rely mainly on non-pharmacological measures. Those measures, especially social distancing, are a challenge for developing countries, such as Brazil. In São Paulo, the most populous state in Brazil (45 million inhabitants), most COVID-19 cases up to April 18th were reported in the Capital and metropolitan area. However, the inner municipalities, where 20 million people live, are also at risk. As governmental authorities discuss the loosening of measures for restricting population mobility, it is urgent to analyse the routes of dispersion of COVID-19 in São Paulo territory. We hypothesize that urban hierarchy is the main responsible for the disease spreading, and we identify the hotspots and the main routes of virus movement from the metropole to the inner state. In this ecological study, we use geographic models of population mobility to check for patterns for the spread of SARS-CoV-2 infection. We identify two patterns based on surveillance data: one by contiguous diffusion from the capital metropolitan area, and other hierarchical with long-distance spread through major highways that connects São Paulo city with cities of regional relevance. This knowledge can provide real-time responses to support public health strategies, optimizing the use of resources in order to minimize disease impact over population and economy.”

3) Introduction

Please describe the hypothesis and objectives. While the merit of the work is recognizable, objectives and hypothesis aren’t clear.

We added the following sentence “Using data since the first confirmed cases of COVID-19 in São Paulo State, we assess the importance of geographic space on the spread of the epidemic. For this, we cross validate the confirmed cases with urban mobility, urban hierarchy, and land use at each spatial localization. The results highlight the importance of the main routes that cross São Paulo State and the regional airports on the introduction of the disease in the territory, just as the main municipalities that act as key centers of disease spreading to inner state. Knowing in advance the path of COVID-19 dispersion can support decision-makers to optimize health service, and plan strategies of quarantine measures.”

4) Methods

4A. Line 46: Add “influence on the predicted value than those far away..” 

Done.

4B. Line 50: Type “April” and the dates described in Abstract and data are different (15th vs 18th of April) given the exponential nature of cases and hospitalizations observed with COVID-19, confirming the correct date would matter. 

Done.

4C. Line 57-59: Please describe how these features were used in the analysis. 

Done.

4D. Line 62: Mention how the SEIR results for 645 municipalities were summarized by the mesoregions it seems (n=18). 

In fact, it was applied only to the cities identified as the hot spot for the disease transmission. 

4E. Line 67: mention that its “..fifteen age groups” and please justify this extensive stratification of age groups with a reference. 

As the contact matrix is obtained using these age groups, so we took into account all of them. If we change the number of age groups, we must combine both information mortality and contact matrix using an average weighted by the size of population in each class of age (lines 193-197).

For the first submission we used parameters from the literature, especially from Wuhan, China. For this second submission we used data from São Paulo, Brazil (https://www.seade.gov.br/) that provide weakly information about the disease. 

4F. Provide relevant reasoning for the initial 10 cases for the age categories 25 - 50 age class.

We added the following sentence (lines 205-207) “Reported cases are shown at the date at which it occurred, but simulations started when ten infected individuals ($t=0$) were confirmed to ensuring that the community transmission was already in course.”

As the infection was introduced by people coming from outside (specially Italy), and it is less transmitted by kids when compared to adults, it is a good assumption that this is the age classes responsible for the coronavirus introduction in Brazil. Besides, the data reinforce this assumption since most of reported cases in the beginning of the epidemic in each city are in these age classes.

4G. Line 88-be more specific what "disease control" entails-does this differ from "social distancing" references in line 75?

Social distancing is a non-pharmacological measure that helps to control disease transmission.

4H. Line 108: Heterogeneity of what characteristic? Age categories?

The contact matrix, the mortality, and population distribution are different.

4I. SEIR model assumptions and definitions are not mentioned explicitly.

Model assumptions take into account the epidemiological course of the disease. This is not really a new model, and we are using it here only to exemplify and discuss how to manage epidemic with non-pharmacological measures that diminishes the contact rate.

4J. Provide a reference for the choice of 2.7 as R0

We change all this part on the text, R0=2.7 was the one measured in Wuhan, China. As coronavirus, at that time, was just introduced in Brazil, R0 was not measured at São Paulo territory yet. In the new analysis, we varied R0 for several values (1.05-2) and compared the results of simulation to the reported number of cases. Please, check lines 204-207 and Figure 6.

4L. Are you assuming the rates of transmission in the inner municipalities is comparable to capital and the metropolitan areas? If yes, please explain why this assumption was made.

We rewrote the discussion since we are almost nine months since the introduction of the virus in national territory and now we know more about disease behaviour. Please check lines 288-331.

4M. Do you assume same parameter values for all age groups? Please mention if this assumption was made. Except for Lines 79-82 most other assumptions seems same for all age groups

No, the mortality depends on the age class. Other heterogeneities are the number of individuals in each age class and the contact among individuals in different age classes.

4N. It is unclear how the contiguity, primary and secondary axis were defined when interpreting SEIR model results (Table 2 and Fig 3). Include details on the definitions and please describe. 

The section “Results and Discussion” was rewrote to clarify all the comments done here. In particular, new figures were added and the time that they appear on the text was rethought. Also, a huge improvement was done in the section “Methods”.

4O. The definitions of axis of dispersion along the major locomotion routes may involve assigning a relative time connection matrix to recognize the average direction of the disease spread over time.

We rewrote it. Please check lines 112-127.

4P. What software/s were used to perform the analysis and illustrate?

We used ArcGIS 10.8. We added this information on the manuscript.

5) Results & Discussion

5A. Line 92-94: Please revise the sentence. Typo: “de” 

Done.

5B. Line 100: “peripheral” municipalities in lieu of ‘pole’ municipalities 

Done.

5C. Explain the details depicted in Fig 2 in detail in the text. 

Done.

5D. Please explain the key limitations related to the data. The reference 11 is the database it seems and it is unclear to the reader what are the limitations. 

The data was obtained from Brasil IO’s compiled databank (https://brasil.io/dataset/covid19/boletim/) that is keep by task force of volunteers (researchers and reporters). This group daily catches, from the epidemiological reports of each state, the number of confirmed cases and death by COVID-19, and let it publicly available in a website. Laboratory certification in Brazil ranged from few laboratories to 26 labs, in eight weeks since the starting of the epidemic; the majority is at São Paulo State (Grotto, R., et al (2020). Increasing molecular diagnostic capacity and COVID-19 incidence in Brazil. Epidemiology and Infection,148, E178. doi:10.1017/S0950268820001818). Data reporter of Severe Acute Respiratory Illness (SARI) is mandatory in Brazil. A specific form (national database SIVEP-Gripe) collect several information that allows us to estimate reported delay, disease fatality at which age class, and identify confirmed cases of COVID-19. This permits surveillance of respiratory disease on Brazil. Only cases that were hospitalized belong to this data set, therefore, underreporting is expected. On the other hand, this is probably homogeneous along the municipalities and therefore, will not impact on the observed pattern of disease spreading. To detect underreporting a broader serological survey can be done. There is one in course in Brazil that is in its fourth round, but for the propose of this study this data cannot be use, since they are recent. We added some of this information to the text in lines 53-76.

5E. Lines 125 – 126: Please explain how does the pressure from industry and trading companies are affecting the social distancing requirements.

At the moment we were writing the work, in April 2020, there was a pressure from industry and commerce to implement the relaxing measures. Nowadays, almost six months later, the relaxing plan has already started and most of services are allowed to work. This sentence was deleted since it does not have sense in the current scenario of social-distancing measures. 

6. While the recommendation for social distancing and its impact in reducing the disease is widely known, please discuss how this specific analysis translates to inform disease control?

This analysis helps to understand how the COVID-19 epidemics spreads in the São Paulo State, and demonstrates that it does not follow a diffusive pattern, but a hierarchical spread. We added the paragraph in the conclusion pointing this important result of our work. Please, see lines 350-355.

“This work shows that it was possible to understand and even predict the route of COVID-19 spread in São Paulo State looking to the cities' hierarchy, which means that the spread of epidemic does not follows a diffusion process, but reaches the cities based on their regional importance and activities. After that, the epidemic diffusively spread to contiguous cities. We state that those cities are responsible for the arrival of the epidemic in the inner São Paulo State, and demand attention.”

7. What distance from the capital/metropolitan area got highly affected, within what time-frame, and the major two routes of locomotion identified through the analysis as mainly involved in the dispersal, does these routes have specific characteristics?

At that moment (data was from April 18th), in the beginning of epidemic in Sao Paulo state (7th week from the first case) the confirmed cases were already reported in 227 municipalities (35% of total), mainly in areas close to the capital and regional centres. Three weighted standard deviation ellipses are shown on March 29th, April 08th, and April 18th. The angles are 128, 135, and 137 degrees, respectively, and the semi-major axes measures 34, 89, and 110 km. As time pass and epidemic evolves, we can notice a change of direction and velocity of disease spreading. A simple calculus gives us 5.5 km/day and 2.1 km/day. Interesting to note that on March 24th, mobility restriction was imposed ain São Paulo State. 

Tables and figures

Fig 4.

- Font size of the legend need to be increased. 

- Include a scale bar and garticules

Following the suggestion, the former Figure 1 was remade and now is presented as Figure 4, and the caption of it was improved. New caption:

 “Dispersion map for COVID-19 in São Paulo State Brazil from March 25 to April 18, 2020. The color spectrum indicates the areas of early (in red) to those of more recent COVID-19 introduction (in blue). The main roadways that cross São Paulo State, and the urban hierarchy level of each municipalities is displayed.”

Fig 5.

- The figure is too busy with multiple sub figures and no clear legend explaining the map details. For example what does the size of circular symbols represent?

Following the suggestion, Figure 5 was redone and the caption of it improved. New caption: “Elementary spatial structures associated to COVID-19 spread in São Paulo State, Brazil. Around the main map we display the structures that comprise it, such as road axes, regional airports, metropolitan area of São Paulo city, municipalities keys as center of disease dispersion from metropolitan area to inner state, and municipalities where the oldest population of São Paulo lives.”

We also added in the text the following sentence (lines ): “To understand the regional pattern of SARS-CoV-2 spreading, Fig. 5 presents the elementary spatial structures identified as the main responsible for the disease spread inside the state. They comprise the main roadways as well as the airports that give people and trade mobility, and the hotspots of the disease introduction and spread. The airports and the cities are displayed by circles proportional to their role on COVID-19 spread. In the case of cities, we classified them as principal (São Paulo, Campinas, São José dos Campos, Ribeirão Preto, and São José do Rio Preto) and secondary (Santos, Araçaatuba, Presidente Prudente, Bauru, Marília, São Carlos, Sorocaba, Rio Claro, and Piracicaba) urban centers in the level of relevance for the disease spread. The metropolitan area of São Paulo and the region of the state where there is a massive concentration of oldest population (older than 60 years of age) are highlighted. The later is called vulnerable because disease lethality among them is high.”

- Include a scale bar for the main figure. 

This figure (Figure 5) is a schematic figure and must not have a scale.

- “Secondary” typos in two places.

We corrected it.

Fig 5.

- While Table 2 description claims to have defined the variable ‘Connection with the Capital’ based on Fig 3. The figure does not illustrate what areas are considered as ‘Contiguity’, ‘Primary Axis’ and ‘Secondary axis’. Please clarify and change the figure and table labeling.

Now, Table 2 references Figure 5, which shows the axes and metropolitan area, using in the description of it.

We added the sentence in lines 270-273 to clarify the : “Cities belonging to the metropolitan area, such as Santos, São José dos Campos and Campinas, are classified as ``contiguity'', the other ones are connected to São Paulo City through a primary or secondary axis.”

- Define what do you refer to as ‘axis’ in the text. The figure shows two different information and both of these are labelled as ‘axis’

We changed the legend’s title from “axes of virus dispersion” to “Standard deviation ellipses-SDE” and relabelled the legend to “major axis of SDE”. Also, we changed the legend’s title “Main routes of COVID-19 dispersion” to “Main roads”. This way, we believe that any confusion is avoided.

o ‘Axes of virus dispersion’: Standard deviation ellipses has major and minor semi axis and a rotation. Please mention these numerically in a table.

We added the information in lines 216-218: “The angles are 128, 135, and 137 degrees, respectively, and the semi-major axis measures 34, 89, and 110 km.”

o “Main routes of COVID-19 dispersion’ . This sounds synonymous to the previous, except that the features are recognizing the major roads that contribute to the disease spread. It is unclear what does the thickness of the lines represent. Describe and relabel accordingly.

We changed the legend’s title “Main routes of COVID-19 dispersion” to “Main roads”.

Fig 6 and 7.

- Please present them as one figure

- If there were 18 mesoregions in the analysis, as seen in Table 2, why does the graphs show only 17? Why does the figures exclude ‘Registro’?

We changed the figure and table in order to encompass only the cities presented in Figure 5.

We are giving a special attention to cities that are considered important in a regional level. We changed the table and the analysis, which currently encompasses 14 cities, as it is shown in Table 2.

Table 2.

- Column 4: remove the capitalization of ‘Capital’ as it is not consistent with the other column names

Done.

---

## [Decision Letter · Decision Letter 1]

8 Dec 2020

PONE-D-20-12111R1

The use of health geography and compartmental modeling to understand early dispersion of COVID-19 in São Paulo, Brazil.

PLOS ONE

Dear Dr. Fortaleza,

Thank you for submitting your manuscript to PLOS ONE. After careful consideration, we feel that it has merit but does not fully meet PLOS ONE’s publication criteria as it currently stands. Therefore, we invite you to submit a revised version of the manuscript that addresses the points raised during the review process.

We look forward to receiving your revised manuscript.

Kind regards,

Javier Sanchez

Academic Editor

PLOS ONE

Reviewers' comments:

Reviewer's Responses to Questions

**Comments to the Author**

1. If the authors have adequately addressed your comments raised in a previous round of review and you feel that this manuscript is now acceptable for publication, you may indicate that here to bypass the “Comments to the Author” section, enter your conflict of interest statement in the “Confidential to Editor” section, and submit your "Accept" recommendation.

Reviewer #1: All comments have been addressed

Reviewer #2: All comments have been addressed

2. Is the manuscript technically sound, and do the data support the conclusions?

Reviewer #1: Partly

Reviewer #2: Partly

3. Has the statistical analysis been performed appropriately and rigorously? 

Reviewer #1: Yes

Reviewer #2: N/A

4. Have the authors made all data underlying the findings in their manuscript fully available?

Reviewer #1: Yes

Reviewer #2: Yes

5. Is the manuscript presented in an intelligible fashion and written in standard English?

Reviewer #1: Yes

Reviewer #2: Yes

6. Review Comments to the Author

Reviewer #1: The authors conducted a thorough review of the article.

However, regarding the mathematical model, figure 6 clearly demonstrates that the SEIR model, as simulated, is not able to adequately represent the evolution of the disease in the state (points in blue).

This is probably a result of the interventions adopted and that are not implemented in the model. One way to better model social distancing would be to make the beta value variable during the simulation by adjusting it based on the data (capturing the dynamics of social distancing). This would make the simulations more accurate. However, since the model does not contribute to the central hypothesis of the article, I suggest its removal.

Reviewer #2: Authors have made majority of the changes as suggested and have provided appropriate explanations when applicable. Overall the presentation of the paper has improved compared to the previous submission and in a good shape for publication. Minor suggestions are listed below.

Abstract and introduction

Still the hypothesis is not clearly presented. It seems the authors hypothesized a ‘diffusion’ spread and eventually suggested that the hierarchy of the regions matters compared to the contiguity. I suggest to consider revision.

Methods

1. SDE is simply a descriptive representation of the mean center and directionality of the cases at a given time. While authors have put good effort to explain SDE calculation, it is unclear why the three days were selected March 29th, April 8th, and April 18th. Please explain.

2. It is unclear how the airplanes (Fig 2) map was used in the analysis?

3. What software (and what packages; if applicable) were used to model SEIR models. Include references. It is good practice to make the codes available. Or publish as a repository.

4. The description has a major gap that need explanation how SEIR models are using the locomotion directionality information in the models. While the results discuss the flux network it has not been clearly explained in methods.

5. SEIR modes: Basic model description is sufficient however, whether model simulations were run separately and simultaneously for each of the fourteen hotspot cities is unclear.

6. Moreover, as the reviewer #1 has mentioned the models can be validated now that there is more data gathered.

7. SEIR model assumptions and definitions: While SEIR models aren’t new if you’re assuming

- Random mixing vs specific contact rates among individuals in different age classes these need to be stated clearly

- Include the start day for each 14 cities in Table 2

8. Lines 304 – 306. The hypothesis of hierarchy being more important than the contiguity to S~ao

Paulo as a risk factor

Results and discussion

1. The discussion needs a paragraph on limitations including: testing and data dependence, lack of validation, model assumptions, and the while the model helps to understand the early spread of COVID-19 with in a region, without having realtime data at more granular level, the difficulty to assess the risk profile or predict the risk spatiotemporally.

2. The results or interpretations are not discussed in comparison to the recent literature on similar studies. Here are few examples. Authors may find more as there is a plethora of available literature:

- Chen et al. 2020. Controlling urban traffic-one of the useful methods to ensure safety in Wuhan based on COVID-19 outbreak. doi: 10.1016/j.ssci.2020.104938

- Bertuzzo et al., 2020. The geography of COVID-19 spread in Italy and implications for the relaxation of confinement measures. Doi: https://doi.org/10.1038/s41467-020-18050-2

How the information may be helpful for post COVID disease preparedness in cities:

- Pisano C. 2020. Strategies for Post-COVID Cities: AN insight to Paris en commun and Milano Sustainability 2020, 12(15), 5883; https://doi.org/10.3390/su12155883

Figures

Modifications of the figures have improved the overall understanding of the methods and results. However, here are specific comments on critical points:

1. Figure 1: The red circles represent 1, 100, and 5000 cases it seems. Please remove the period (.) from both legend and the map (Sao Paulo 9428). It is misleading

2. Figure 2 Insert a legend onto the figure: Color coded nodes and edges

3. Figure 4: The legend color scheme for “number of days since the first case” does not match what is on the map. Please use the same color scheme/ramp and edit the legend accordingly.

References

Several references have short-form of the journal name. Please edit the references according to Plos guidelines.

7. PLOS authors have the option to publish the peer review history of their article (what does this mean?). If published, this will include your full peer review and any attached files.

Reviewer #1: **Yes: **FERNANDO FERREIRA

Reviewer #2: **Yes: **Kaushi Kanankege

---

## [Author Response · Author response to Decision Letter 1]

19 Dec 2020

Botucatu, December 25th 2020,

Response to reviewers, 

We thank the reviewers for their careful reading and suggestions. Below, we answer the questions, clarifying points, and highlighting the changes made regarding the arisen points.

Reviewer #1: The authors conducted a thorough review of the article.

However, regarding the mathematical model, figure 6 clearly demonstrates that the SEIR model, as simulated, is not able to adequately represent the evolution of the disease in the state (points in blue).

This is probably a result of the interventions adopted and that are not implemented in the model. One way to better model social distancing would be to make the beta value variable during the simulation by adjusting it based on the data (capturing the dynamics of social distancing). This would make the simulations more accurate. However, since the model does not contribute to the central hypothesis of the article, I suggest its removal.

We thank the comment, and we agree entirely with it. Perform estimation for different betas throughout the time would diverge from the work goal and its scope; therefore, we opted for the removal of the mathematical model from the text.

Reviewer #2: Authors have made majority of the changes as suggested and have provided appropriate explanations when applicable. Overall the presentation of the paper has improved compared to the previous submission and in a good shape for publication. Minor suggestions are listed below.

Abstract and introduction

Still the hypothesis is not clearly presented. It seems the authors hypothesized a 'diffusion' spread and eventually suggested that the hierarchy of the regions matters compared to the contiguity. I suggest to consider revision.

We rewrote the last paragraph of the introduction to make it clear (lines 23-40). We also change the word "diffusion" to "dispersion" to avoid misunderstanding. We hope that it is clear now. In red, the changes are highlighted. "Here, we discussed a detailed analysis of the spatial dispersion of COVID-19 in São Paulo State, Brazil, intending to provide real-time responses to support public health strategies. Using data since the first confirmed cases of COVID-19 in São Paulo State, we assess the importance of geographic space on the spread of the epidemic. We hypothesize that urban hierarchy is the main responsible for the disease spreading, and we identify the hotspots and the main routes of virus movement from the metropolis to the inner state. This premise is also supported by [8] where multivariate analyses showed that demographic density and high classification of regional relevance were associated with early introduction and high COVID-19 incidence and mortality rates. In work developed here, we cross-validate the confirmed cases with urban mobility, urban hierarchy, and land use at each spatial localization. The results highlight the importance of the main routes that cross São Paulo State and the regional airports on the introduction of the disease in the territory, just as the main municipalities that act as key centers of disease spreading to the inner state. Knowing in advance the path of COVID-19 dispersion can support decision-makers to optimize health service and plan strategies of quarantine measures. This approach can be done in other states of Brazil and other developing countries, observing local and regional differences of mobility and urban network."

In the abstract of the first revision version the sentence "We hypothesize that urban hierarchy is the main responsible for the disease spreading, and we identify the hotspots and the main routes of virus movement from the metropolis to the inner state" was already added.

 The following reference was added:

[8] Fortaleza CMCB, Guimarães RB, de Almeida GB, Pronunciate M, Ferreira CP (2020). Taking the inner route: spatial and demographic factors affecting vulnerability to COVID-19 among 604 cities from inner São Paulo State, Brazil. Epidemiology and Infection 148, e118, 1–5. https://doi.org/10.1017/ S095026882000134X

Methods

1. SDE is simply a descriptive representation of the mean center and directionality of the cases at a given time. While authors have put good effort to explain SDE calculation, it is unclear why the three days were selected March 29th, April 8th, and April 18th. Please explain.

We added the following sentence to justify the calendar days used (lines 121-130): "Although the SARS-CoV-2 was introduced in São Paulo on March 25th, it took time to move from it to the inner municipalities because of the strong mitigations strategy adopted by São Paulo State to halting the disease's spread. The average time spent by the disease, since its introduction on the metropolis, to achieve the regional centers, the municipalities under major and minor influence, and the rural municipalities were respectively 22, 31, 34, and 55 days [8] (the classification of the municipalities follows the criteria established by the Brazilian Institute for Geography and Statistics (2017) [17]). Therefore, three calendar dates were chosen to cover the study period (from March 25th until April 18th): March 29th, April 8th, and April 18th, which are 10 days apart from each other."

2. It is unclear how the airplanes (Fig 2) map was used in the analysis?

Fig. 2 emphasizes the movement restrictions implemented during the social distancing reflected in the number of people traveling in São Paulo State, but also highlight that it took time to occur, and this delay helps on disease spreading among the municipalities giving the mix patter observed for the dispersion of the disease: hierarchy + contiguity. We added this figure to answer one of the reviewers' comments that the observed pattern was trivial and expected, i.e., by the main railways. This figure is also crucial for figure 5 that contain the main airports involved in the disease transmission pattern. We added the following sentence in the section results and discussion (lines 220-225): "People's movement is facilitated, and encouraged, due to transportation availability and commercial and social activities Likely SARS-CoV in 2003, the SARS-CoV-2 fastly spread among cities and countries due to airline network and ground transportation [22-24]. In the case of São Paulo State, the delay on closing the airports located at the inner municipalities probably contributed to the hierarchical dispersion of the disease on its territorial."

We added the following references:

[22] Litaker, J. R., Chou, J. Y., Novak, S., & Wilson, J. P. (2003). Implications of SARS: Medical geography and surveillance in disease detection. Annals of Pharmacotherapy,37(12), 1841-1848.

[23] Bowen Jr, J. T., & Laroe, C. (2006). Airline networks and the international diffusion of severe acute respiratory syndrome (SARS).Geographical Journal,172(2), 130-144.

[24] Ali, S. H., & Keil, R. (Eds.). (2011).Networked disease: Emerging infections in the global city (Vol. 44). John Wiley & Sons.

3. What software (and what packages; if applicable) were used to model SEIR models. Include references. It is good practice to make the codes available. Or publish as a repository.

It was implemented in language C using a Runge-Kutta method of 4th order. However, the mathematical model was removed from the text, as suggested by Reviewer 1.

4. The description has a major gap that need explanation how SEIR models are using the locomotion directionality information in the models. While the results discuss the flux network it has not been clearly explained in methods.

The mathematical model was removed from the text, as suggested by Reviewer 1.

5. SEIR modes: Basic model description is sufficient however, whether model simulations were run separately and simultaneously for each of the fourteen hotspot cities is unclear.

They were run separately. However, the mathematical model was removed from the text, as suggested by Reviewer 1.

6. Moreover, as the reviewer #1 has mentioned the models can be validated now that there is more data gathered.

The mathematical model was removed from the text, as suggested by Reviewer 1.

7. SEIR model assumptions and definitions: While SEIR models aren't new if you're assuming

- Random mixing vs specific contact rates among individuals in different age classes these need to be stated clearly

The mathematical model was removed from the text, as suggested by Reviewer 1. Only to clarify, we were using a contact matrix to simulate the difference in the number of contact among individuals from different age classes.

- Include the start day for each 14 cities in Table 2.

Done.

8. Lines 304 – 306. The hypothesis of hierarchy being more important than the contiguity to São

Paulo as a risk factor.

We took out this sentence from the text since it was related to a discussion based on the mathematical model that was deleted from this new version as suggested by Reviewer 1.

Results and discussion

1. The discussion needs a paragraph on limitations including: testing and data dependence, lack of validation, model assumptions, and the while the model helps to understand the early spread of COVID-19 with in a region, without having real-time data at more granular level, the difficulty to assess the risk profile or predict the risk spatiotemporally.

We added two paragraphs in the Discussion section to describe the limitations of the study (lines 309-328): "Limitations of the analysis include: (i) the no-identification of asymptomatic individuals and, potentially, mild or moderate infectious, since only symptomatic cases that seek for medical assistance have been tested; (ii) data dependence, i.e., data set does not distinguish between imported and autochthonous cases; (iii) the assumption that all individuals have the same degree of susceptibility and transmissibility of the disease, regardless the environment they live; (iv) the transmission is homogeneous within the cities; (v) mitigation strategies are the same everywhere. All those characteristics may variate according to the city because the number of tests that is distributed and performed among cities is not homogeneous; the number of contacts among people changes according to the city characteristics, such as the use of public transportation [29]; and people's adherence to social distancing really differed across the state, which may be related to the epidemics delay into reach the small inner cities, affecting people's risk perception [30].

Moreover, the data source in Brazil has been updated with some delay, regarding the occurrence of the infections [31]. Nevertheless, since the data used in this study is related to the arrival of infections in each city, which happened in early 2020, we expect the numbers to be trustful at the point of the analysis. Despite there is no data at a granular level, such as information about the address of the infection occurrence, the data is enough to perform the analysis and reach our goal, which was to study the spread of SARS-CoV-2 among cities."

The following references were added:

[29] Chen et al. 2020. Controlling urban traffic-one of the useful methods to ensure safety in Wuhan based on COVID-19 outbreak. doi: 10.1016/j.ssci.2020.104938

[30] Baicker et al 2020. Using social and behavioural science to support COVID-19 pandemic esponse. Nature Human Behaviour; p. 1--12.

[31] Do Prado et al (2020). Analysis of COVID-19 under-reporting in Brazil. Revista Brasileira de terapia intensiva,32(2), 224.

2. The results or interpretations are not discussed in comparison to the recent literature on similar studies. Here are few examples. Authors may find more as there is a plethora of available literature:

- Chen et al. 2020. Controlling urban traffic-one of the useful methods to ensure safety in Wuhan based on COVID-19 outbreak. doi: 10.1016/j.ssci.2020.104938

- Bertuzzo et al., 2020. The geography of COVID-19 spread in Italy and implications for the relaxation of confinement measures. Doi: https://doi.org/10.1038/s41467-020-18050-2

How the information may be helpful for post COVID disease preparedness in cities:

- Pisano C. 2020. Strategies for Post-COVID Cities: AN insight to Paris en commun and Milano Sustainability 2020, 12(15), 5883; https://doi.org/10.3390/su12155883

We thank the suggestion of the reviewer. To address this point, we added two paragraphs to the discussion section:

(lines 220-225): "People's movement is facilitated, and encouraged, due to transportation availability and commercial and social activities Likely SARS-CoV in 2003, the SARS-CoV-2 fastly spread among cities and countries due to airline network and ground transportation [22-24]. In the case of São Paulo State, the delay on closing the airports located at the inner municipalities probably contributed to the hierarchical dispersion of the disease on its territorial."

(lines 274-284): "A relationship between disease spreading and territorial geography was also established in other epidemics [22,25]. Differently from São Paulo State, [26] showed that the first wave of SARS-CoV-2 pandemic in Germany followed a dispersion pattern called relocation diffusion process since the arrival of infections in Germany coincided with a traditional carnival festivity. Therefore, a single infected individual transmitted the infection to several others. After the festivities, people went back to their homeland, creating long-range connections, and new spots of infection spread, which were randomly distributed across the country. In São Paulo State, since all non-essential activity was limited, the spread followed the main routes of commercial relationships and supply distribution, in a hierarchical diffusion, firstly reaching the most important cities in São Paulo State, and locally spreading within their regions."

And we added the new references:

[22] Litaker, J. R., Chou, J. Y., Novak, S., & Wilson, J. P. (2003). Implications of SARS: Medical geography and surveillance in disease detection.Annals of Pharmacotherapy,37(12), 1841-1848.

[23]Bowen Jr, J. T., & Laroe, C. (2006). Airline networks and the international diffusion of severe acute respiratory syndrome (SARS).Geographical Journal,172(2), 130-144.

[24]Ali, S. H., & Keil, R. (Eds.). (2011).Networked disease: Emerging infections in the global city (Vol. 44). John Wiley & Sons.

[25] Kuo, C. L., & Fukui, H. (2007). Geographical structures and the cholera epidemic in modern Japan: Fukushima prefecture in 1882 and 1895.International Journal of Health Geographics,6(1), 25.

[26]Kuebart, A., & Stabler, M. (2020). Infectious diseases as socio‐spatial processes: The Covid‐19 outbreak in Germany.Tijdschrift voor economische en sociale geografie,111(3), 482-496.

Figures

Modifications of the figures have improved the overall understanding of the methods and results. However, here are specific comments on critical points:

1. Figure 1: The red circles represent 1, 100, and 5000 cases it seems. Please remove the period (.) from both legend and the map (Sao Paulo 9428). It is misleading

Done.

2. Figure 2 Insert a legend onto the figure: Color coded nodes and edges

Done.

3. Figure 4: The legend color scheme for "number of days since the first case" does not match what is on the map. Please use the same color scheme/ramp and edit the legend accordingly.

Done.

References

Several references have short-form of the journal name. Please edit the references according to Plos guidelines.

Done.

---

## [Editor Report · Decision Letter 2]

22 Dec 2020

The use of health geography and compartmental modeling to understand early dispersion of COVID-19 in São Paulo, Brazil.

PONE-D-20-12111R2

Dear Dr. Fortaleza,

We’re pleased to inform you that your manuscript has been judged scientifically suitable for publication and will be formally accepted for publication once it meets all outstanding technical requirements.

Kind regards,

Javier Sanchez

Academic Editor

PLOS ONE
---

## [Editor Report · Acceptance letter]

29 Dec 2020

PONE-D-20-12111R2 

The use of health geography modeling to understand early dispersion of COVID-19 in São Paulo, Brazil 

Dear Dr. Fortaleza:

I'm pleased to inform you that your manuscript has been deemed suitable for publication in PLOS ONE. Congratulations! Your manuscript is now with our production department. 

Kind regards, 

on behalf of

Dr Javier Sanchez 

Academic Editor

PLOS ONE